# Robust projection of East Asian summer monsoon rainfall based on dynamical modes of variability

Daokai Xue [1] ✉, Jian Lu [2] ✉, L. Ruby Leung [2], Haiyan Teng[3], Fengfei Song [4,5], Tianjun Zhou [6] & Yaocun Zhang[1]

The Asian monsoon provides the freshwater that a large population in Asia depends on, but how anthropogenic climate warming may alter this key water source remains unclear. This is partly due to the prevailing point-wise assessment of climate projections, even though climate change patterns are inherently organized by dynamics intrinsic to the climate system. Here, we assess the future changes in the East Asian summer monsoon precipitation by projecting the precipitation from several large ensemble simulations and CMIP6 simulations onto the two leading dynamical modes of internal variability. The result shows a remarkable agreement among the ensembles on the increasing trends and the increasing daily variability in both dynamical modes, with the projection pattern emerging as early as the late 2030 s. The increase of the daily variability of the modes heralds more monsoon-related hydrological extremes over some identifiable East Asian regions in the coming decades.

Asian monsoon affects a large area of South and East Asia, which are home to almost half of the world's population. How the Asian monsoon rainfall is projected to change under anthropogenic climate warming has profound social and ecological consequences for the Asian populations and is of critical importance to the stakeholders and policy makers in these Asian countries. Considerable efforts have been made to project the future monsoon changes under global warming. While the increased water vapor holding capacity of the warming atmosphere will increase the summer monsoon precipitation in general[1–3], how the monsoon circulation will respond to warming has been mainly construed to be the source of the uncertainty[4–8]. Meanwhile, evidence began to emerge for the intensification of the Northern Hemisphere (NH) summer monsoon, attributable to the faster warming rate in the NH compared to the Southern Hemisphere under global warming[9]. However, uncertainties in the projection of the regional monsoons[10–13] remain large, not least due to the specific dynamic and thermodynamic components and their complex interactions for each regional monsoon system. Furthermore, model biases in representing the rainfall and circulation of the regional monsoon systems persist across generations of climate models, undermining the fidelity in the projections made by climate models. For example, climate models still have difficulty in simulating both the mean position and the variability of the Meiyu-Changma-Baiu rainbelt over eastern China, Korean Peninsula, and southern Japan[14–19]. East Asia and the Indian subcontinent are among the regions with the largest model uncertainties in the end-of-the-century precipitation projection[12,20–23]. As society demands reliable climate change information for the purpose of planning, adaptation, and policy making, any increase in confidence in regional climate projections can be of great societal value.

As astutely stated in the seminal work of K. Hasselmann[24], climate projection is a problem of "fingerprinting", that is, capturing the forced response in a much-reduced dimension through pattern

[1]School of Atmospheric Sciences, Nanjing University, Nanjing, China. [2]Atmospheric Sciences and Global Change Division, Pacific Northwest National Laboratory, Richland, WA, USA. [3]Climate and Ecosystem Sciences Division, Lawrence Berkeley National Laboratory, Berkeley, CA, USA. [4]Frontier Science Center for Deep Ocean Multispheres and Earth System and Physical Oceanography Laboratory, Ocean University of China, Qingdao, China. [5]Laoshan Laboratory, Qingdao, China. [6]State Key Laboratory of Numerical Modeling for Atmospheric Sciences and Geophysical Fluid Dynamics, Institute of Atmospheric Physics, Chinese Academy of Sciences, Being, China. ✉e-mail: dkxue@nju.edu.cn; jian.lu@pnnl.gov

filtering so as to greatly improve the signal-to-noise ratio. In the same vein as Hasselmann and a number of similar studies[25–27], we use the neutral modes of the Asian monsoon rainfall variability as the a priori fingerprints to filter through the model-simulated precipitation signals to provide a more confident projection of regional monsoon rainfall. Although, both serve to reduce the dimension of the signal detection problem, compared to the empirical orthogonal functions (EOFs) originally proposed in ref. 24, which are empirically determined, neutral modes represent the coherent spatial patterns that are dynamically organized and ranked according to the magnitude of the singular values, with the leading mode representing the least damped, and thus most excitable pattern in the climate system (See Methods for the derivation and explanation of neutral modes). In addition, neutral modes capture the more deterministic aspect of the system governed by the robust dynamics of the climate system[28–30]. Therefore, the precipitation patterns extracted through the neutral mode analysis are more robust to climate noise and model uncertainty, and thus more detectable and predictable than the precipitation at a single grid point. Furthermore, the confidence/ robustness of the Asian monsoon rainfall response expressed in neutral modes can be stratified by mode rank, with leading modes being more robust and the confidence derived from them potentially transferable to the real climate change response.

As it turns out, our analysis reveals a consensus on the sign of the change of the leading modes of the East Asian Monsoon (EAM) precipitation across the multi-model ensembles examined here. In addition, the daily standard deviation (std) of the modes is projected to increase at an even greater rate, indicating more extreme daily variations of these modes in a warmer climate. Interestingly, the circulation of the modes contributes more to the increase in EAM precipitation modes than the thermodynamic factors. Within the multi-model ensembles, we also assess the time of emergence (ToE) of the forced changes of these two modes and infer the emergent robust

component of the EAM rainfall response under an uncurbed emission scenario.

## Results

### The leading neutral modes and their trends in CESM-SMILEs

The precipitation dynamic modes are extracted as the leading singular modes of the linear response function (LRF) for precipitation constructed based on Green's function perturbation experiments with the National Center for Atmospheric Research's slab-coupled Community Atmospheric Model version 5 (CAM5-SOM). They are herein referred to as the neutral modes (NMs) of precipitation. The first neutral mode (NM1) for JJA precipitation is shown in Fig. 1a. Over the Asia-Western Pacific sector, NM1 is well identified with the leading EOF pattern (Fig. 1b, c), revealing the iconic tripolar pattern of the EAM rainfall, which features a positive anomaly over the middle and lower reaches of the Yangtze River valley, the southern Korean peninsula, and southern Japan, sandwiched by negative anomalies on both flanks (Fig. 1c). Thus, the tripolar pattern can be interpreted as a part of the global precipitation mode. Similarly, NM2 can be identified with the second EOF of the Asian monsoon precipitation, which is characterized by a dipole, referred to in the literature as the South Flood-North Drought (SFND) pattern[31–36] (Fig. S1). From this point on, the regional imprints of the leading NMs within the domain [85°–150°E, 5°–45°N] will be used as the fingerprints for future projection of the summer rainfall over the East Asian regions.

To this end, CESM2 large ensemble (CESM-LENS2) under the CMIP6 historical forcing and the SSP370 forcing scenario[37] and the large ensembles of four models from the Single-Model Initial-condition Large Ensembles project (SMILEs)[38] under the standard CMIP5 "historical" and "future" Representative Concentration Pathway 8.5 (RCP8.5) forcing protocols are used. These five ensembles are collectively referred to as CESM-SMILEs for short. Due to the different forcing protocols between SSP370 and RCP8.5[39], the trend component

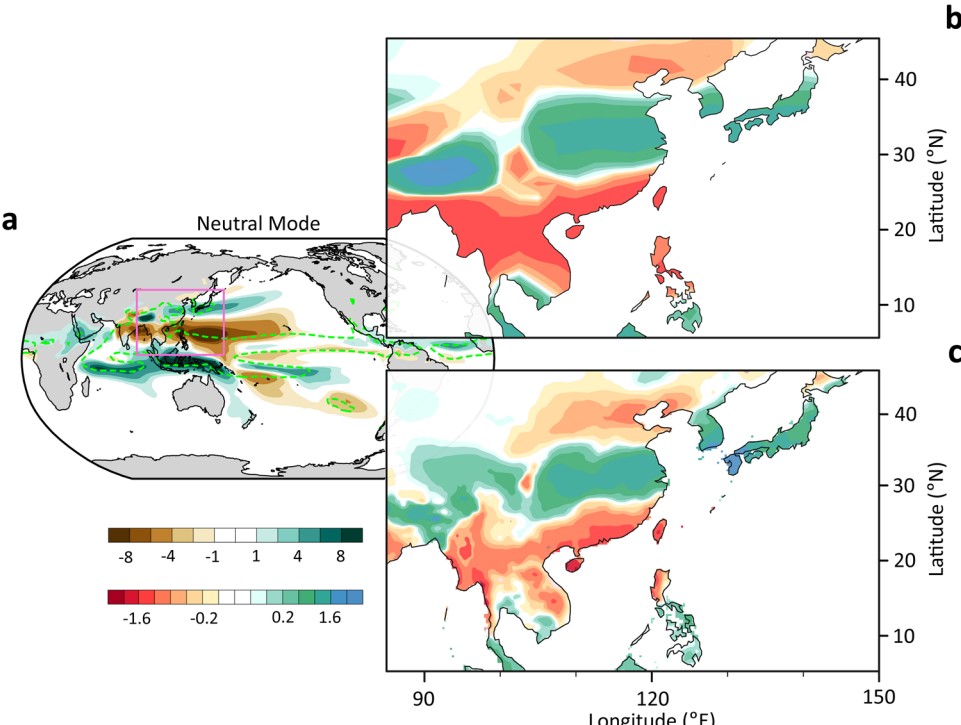

**Fig. 1 | Agreement between the 1st neutral mode (NM1) and the first empirical orthogonal function (EOF1) of East Asian summer monsoon rainfall. a** NM1 of the global June–July–August (JJA) precipitation extracted from the precipitation linear response function (LRF) (unit is mm day⁻¹). The green dashed contours indicate the 6 mm day⁻¹ isoline of the JJA summer rainfall climatology. **b** A close-up of NM1 over East Asia to facilitate comparison with the observed EOF1 pattern of summer rainfall over the same region shown in **c**.

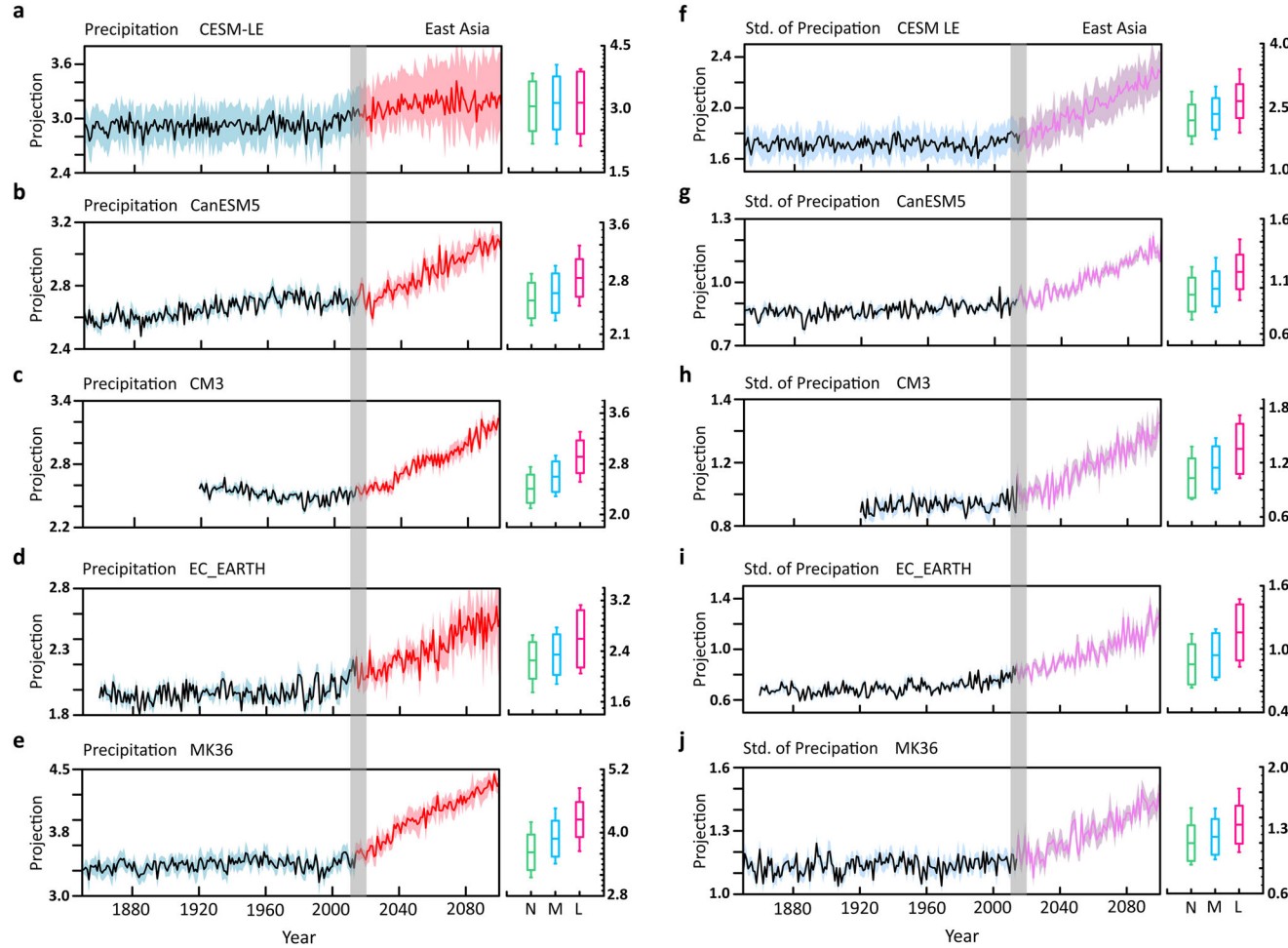

**Fig. 2 | The evolution of the first neutral mode (NM1) projections in CESM-SMILEs.** Left panels: projections of the June–July–August (JJA) seasonal mean precipitation onto the NM1 pattern over East Asia, or the NM1 index. Right panels: The std of summer daily NM1 index. Solid line in each left (right) panel is the ensemble mean projection (ensemble mean of the std of the daily NM1 index). The shading indicates the range of the one std of the ensemble members of each individual ensemble. A gray bar in each panel demarcates the boundary between the "historical" and "future" periods of the simulations. The side box in each panel shows the mean and spread (represented by one std) of the near-term (2021–2040, green), mid-term (2041–2060, blue), and long-term (2080–2100, red) projections. Note the different scales shown on the y axis of each panel.

during the "future" projection period in the CESM-LENS2 ensemble is scaled up by a factor of 1.3 to account for the scenario difference when we estimate the ToE and attribute projection uncertainties (see Methods for the rationale for the scaling factor).

We project the JJA precipitation from CESM-SMILEs onto the standardized NM1 pattern of CESM-LENS2 (since NMs are only available for CESM-LENS2 from its LRF) and the resulting dimensional NM1 indices are shown in Fig. 2. There emerges a consensus on the positive trend in the projected changes in the NM1 index, implying a moistening trend over the corridor from the middle and lower reaches of the Yangtze River valley eastward to southern Japan, and a drying trend over Indochina due to the contribution of NM1. More remarkably, the increasing trend in the JJA std of the daily NM1 index is even greater than that in the seasonal NM1 index, with the grand ensemble mean increasing by 39.8% for the former and 29.2% for the latter by the end of the century (taking into account the scenario differences between CESM-LENS2 and SMILEs). For reasons yet to be understood, the CESM-LENS2 has a much larger internal variability in the NM indices than other ensembles (Fig. S2). Unanimous agreement is also found on the projections of the JJA mean and std of the NM2 index across the CESM-SMILEs ensembles, with the grand ensemble mean NM2 index increasing by 26.5% and the grand ensemble mean std of the NM2 index increasing by 32.7% by the end of the century. Treating the

CESM-SMILEs as a grand ensemble, we will show that the grand ensemble mean trends in both the NM1 and NM2 indices are significant relative to the internal noise when we examine the ToE of the forced signals. Over the domain shown in Fig. 1b, NM1, and NM2 together explain more than 50% of the total spatial variance in the June–July–August (JJA) seasonal mean precipitation trend between 2011 and 2100. Notably, since NM2 represents the negative SFND pattern (Fig. S1), a positive trend in the NM2 index in the future (Fig. S3e) would mitigate the positive SFND trend over the past six decades in China.

### Uncertainty attribution and time of emergence

Following Lehner et al.[20], we partition the sources of uncertainty into the component due to the internal variability uncertainty and the component due to model uncertainty as follows

$$T_t = \bar{\bar{I}}_t + M_t \qquad (1)$$

each estimated as the variance for a given summer (denoted by subscript $t$) for each NM index. The fractional uncertainty for the two sources at a given year $t$ can be simply calculated as $\frac{M_t}{T_t}$ and $\frac{\bar{\bar{I}}_t}{T_t}$, respectively. See Methods for the detailed definitions. Since the

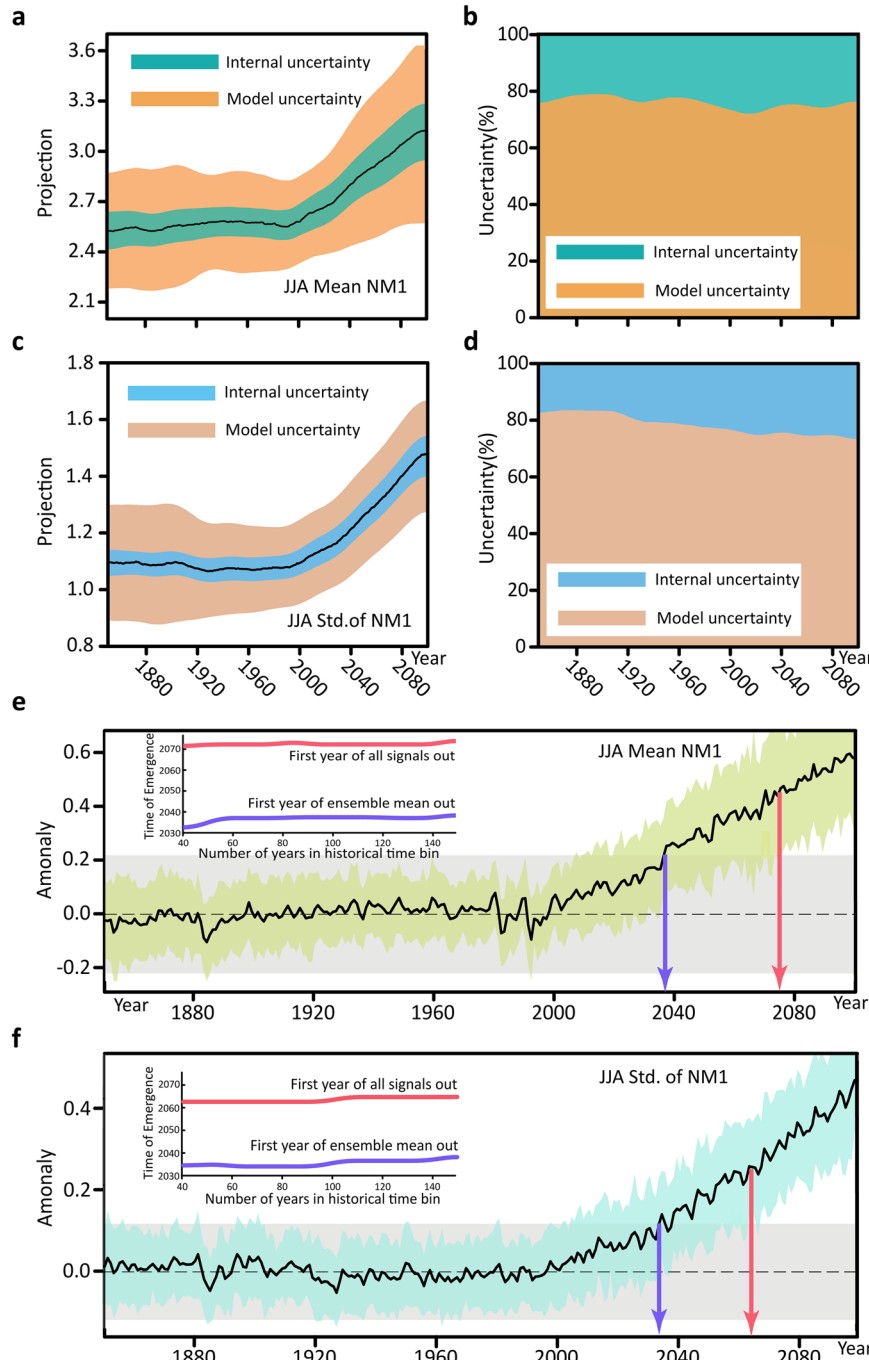

**Fig. 3 | The evolution of uncertainty components and time of emergence (ToE) of the forced response. a** Evolution of the multi-model ensemble mean (MMEM) first neutral mode (NM1) index and its internal variability and model uncertainties represented by two std (i.e., $2\sqrt{\bar{i}_t}$ and $2\sqrt{M_t}$, respectively) on either side of the MMEM; **b** fractional contribution to the total uncertainty from the two uncertainty sources; **c** same as **a** but for the std of the NM1 index; **d** same as **b** but for the std of the NM1 index. Note that a 10-year running mean has been applied to these time series. **e** Time evolution of the forced response of the NM1 index (black line) with respect to the historical (1850–2010) mean. The light green shading indicates the upper and lower bounds of the grand ensemble. The blue and red arrows delineate the ToE based on two definitions (see text for details). The corresponding sensitivities to the length of the reference time window are also depicted in the inset. **f** is the same as **e**, but for the std of the NM1 index. In both **e** and **f**, the gray shading indicates the upper and lower bounds of the grand internal variability during the historical period (1850–2010).

SMILEs ensemble simulations are all forced by the same RCP8.5 scenario, and the end-of-the-century forcing for the CESM-LENS2 has been rescaled to be 8.5 Wm$^{-2}$, scenario uncertainty can be ignored here.

Figure 3a shows the multi-model ensemble mean (MMEM) evolution of the NM1 index (black line) together with its model uncertainty (orange shading) and internal variability uncertainty (green shading). It

is interesting to note that from the turn of the century to the end of the 21st century, both model and internal uncertainties increase with time (so a hinderance to the early detection of the NM1 index trend), while their fractional contributions to the total uncertainty remain roughly unchanged (Fig. 3b). However, a distinct behavior is observed in the evolution of the uncertainties of the NM2 index, with the contribution of model uncertainty increasing during the second half of the 20th

century and decreasing during the 21st century (Fig. S3). While one might expect that increasing water vapor holding capacity with climate warming may increase the variance of the tropical precipitation, which can inflate the uncertainties, the different behavior between the two modes suggests that more complex factors may be at play behind the time dependence of the neutral mode uncertainties. In contrast to the seasonal NM1 index, the model uncertainty for the std of the daily NM1 index shrinks during the 20th century and then stabilizes during the 21st century (Fig. 3c). Thus, in a fractional sense, the internal variability uncertainty in the NM1 std increases consistently with time over the period considered (Fig. 3d). The same is true for the NM2 std (Fig. S3d). Compared to the seasonal mean NM1 index, the relatively stable total uncertainty and the larger forced trend in the std of the NM1 index imply a higher detectability for the change in the latter (cf. Fig. 3a, c). As the std here reflects the behavior of the hydrological extremes within the monsoon system, more process-oriented investigations in this regard are warranted in the future. The two-component partitioning of the uncertainty should be distinguished from the three-component partitioning reported in the literature[20,40], which often shows a decreasing trend in the fractional contribution from the internal variability uncertainty due to the rapidly diverging scenarios after the mid-21st century.

When will these upward trends in the NM indices and their std be detectable in the 21st century? We answer this question by identifying the ToE of the forced changes. Two definitions of ToE are considered: one is more stringent, defined as the year at which the lower bound of the NM indices first exceeds the upper bound of the historical NM indices in the grand ensemble; the other is more lenient, defined as the year at which the grand ensemble mean of the NM indices first exceeds the upper bound of the historical NM indices. The former ToE is also known as "first year of all signals out" and the latter one as "first year of ensemble mean out". The upper and lower bounds of the historical indices (indicated by the gray area in Fig. 3e, f) are identified from the collection of the inter-member spreads over the period 1850–2010 from each of the five individual ensembles. The "first year of all signals out" is detected by year 2073 for the NM1 index and by year 2065 for the NM1 std (Fig. 3e, f), while the "first year of ensemble mean out" is detected much earlier, before 2040 for both the seasonal NM1 index and the NM1 std. Fig. 3e, f also show that the ToEs are relatively insensitive to the historical bin size used to define the lower and upper bounds of the internal variability (see the insets). Because the forced trends in NM2 are relatively weak, only the lenient ToE can be detected within this century (Fig. S3e and S3f). It should be noted that the ToE can differ substantially when assessed separately for each individual ensemble. For example, the stringent ToE cannot be detected within the century for the NM1 index in CESM-LENS2 due to its large internal variability and relatively weak trend (see Fig. S5).

## Dynamic contribution to the NM trends

Even in the absence of circulation change, increasing moisture in the atmosphere can enhance the existing pattern of the hydrological cycle, including the patterns of the leading modes of variability[41]. Does the circulation change play any role in the forced NM trends above, or are they purely thermodynamically driven? To infer the dynamical factors, we first compute the patterns associated with NM1 for SST, 850 hPa wind, 500 hPa geopotential height, specific humidity, and streamfunction (Fig. 4a, b; see Methods for the calculation of the NM associated patterns). The resulting patterns resemble very much those of the well-known 'Pacific-Japan pattern' in the literature[42,43], suggesting that the Pacific-Japan pattern is the manifestation of the leading neutral mode of JJA precipitation over the East Asia-Pacific sector[31]. We then project the 850 hPa streamfunction ($\psi$850) and specific humidity ($Q$850) anomalies onto their corresponding NM1-associated patterns to elucidate the evolution of the dynamic and thermodynamic components of NM1, respectively. In so doing, we assume that the NM1-

associated pattern in Q850 represents the thermodynamic fingerprint of the NM1 circulation on the background mean moisture field. There is a clear upward trend during the "future" period in both the dynamic and thermodynamic projection coefficients (Fig. 4c, d). Specifically, the fractional increase of the dynamic trend is ~19.6% relative to the "historical" mean, accounting for 67% of the total trend of the NM1 index (Fig. 4e). This leaves only 33% for the thermodynamic cause; this figure is somewhat corroborated by the 13.8% fractional increase in the Q850 trend (Fig. 4d). Thus, the circulation changes over the East Asia-Pacific sector contribute qualitatively more to the precipitation changes associated with NM1 than the thermodynamic factor[44,45]. This is also true for the NM2 trend, where the inferred dynamical factor contributes 58% and the thermodynamic factor 42% to the total increase of the NM2 precipitation (see Fig. S4c–e). In addition to the significant contribution of the dynamic factor to the ensemble mean increase in the precipitation modes, the large uncertainty in the NM circulation pattern (shading in Fig. 4c and Fig. S4c) suggests that the dynamical factor also dominates the uncertainty in the future projection of these neutral modes, consistent with the established notion on the source of the uncertainty in regional precipitation projections[7,46,47].

## Projection from a synthetic ensemble

Since each of the five large ensembles provides projections for the NM1 and NM2 indices, one can form 25 synthetic projections for the end-of-the-century (2081–2100) JJA precipitation over East Asia by randomly pairing the projections of the NM1 and NM2 indices from the 5 ensembles as follows

$$P_{i,j} = c_{1,i}V_1 + c_{2,j}V_2 \qquad (2)$$

where $c_1$ and $c_2$ are the dimensional NM1 and NM2 projection coefficients, and $V_1$ and $V_2$ are the non-dimensional patterns of NM1 and NM2, respectively, and the subscripts $i$ and $j$ index the 5 models. This exercise is further motivated by the significant pattern correlations between the EOF1s/EOF2s of the CESM-SMILEs models and the observation (see Table S1). As shown in Fig. 1 and Fig. S1, the NM1/NM2 patterns here are representative of the mode behavior in both climate models and reality. Figure 5a shows the synthetic ensemble mean projection of the East and Southeast Asian JJA precipitation by the end of the century, along with colored dots illustrating the >90% agreement of the 25 synthetic projections on the sign of the precipitation change. Through the lens of NMs, there is considerable consensus on the increase in summer precipitation over the middle and lower reaches of the Yangtze River. This wet area extends north to the Yellow River basin and then west to the Gobi Desert, and east to South Korea and southern Japan. A robust moistening also emerges over Indochina and the northern Philippines. On the other hand, the southern provinces of China including Tibet should expect a drier-than-normal summer climate by the end of the century. In addition, compared to the conventional projecting using multi-model ensemble mean (Fig. 5b), the regional agreement among the synthetic ensembles is much greater.

A similar synthetic ensemble can also be constructed by projecting the 21st century JJA precipitation trends simulated by 36 CMIP6 climate models onto the two leading NMs. The resulting synthetic ensemble mean projection and the 90% consensus across its 1296 members (Fig. 5c) are in very good agreement with the CESM-SMILEs-based ensemble, although with somewhat weaker magnitude and shrunk areas of consensus as more diverse models are included. It is important to note that the NMs-based projections yield different patterns from those of the conventional multi-model ensemble mean projection, the latter being instead characterized by a negative SFND pattern in eastern China with little inter-model consensus (compare the NM-based projections in Fig. 5a, c with their corresponding conventional projections in Fig. 5b, d). We also note

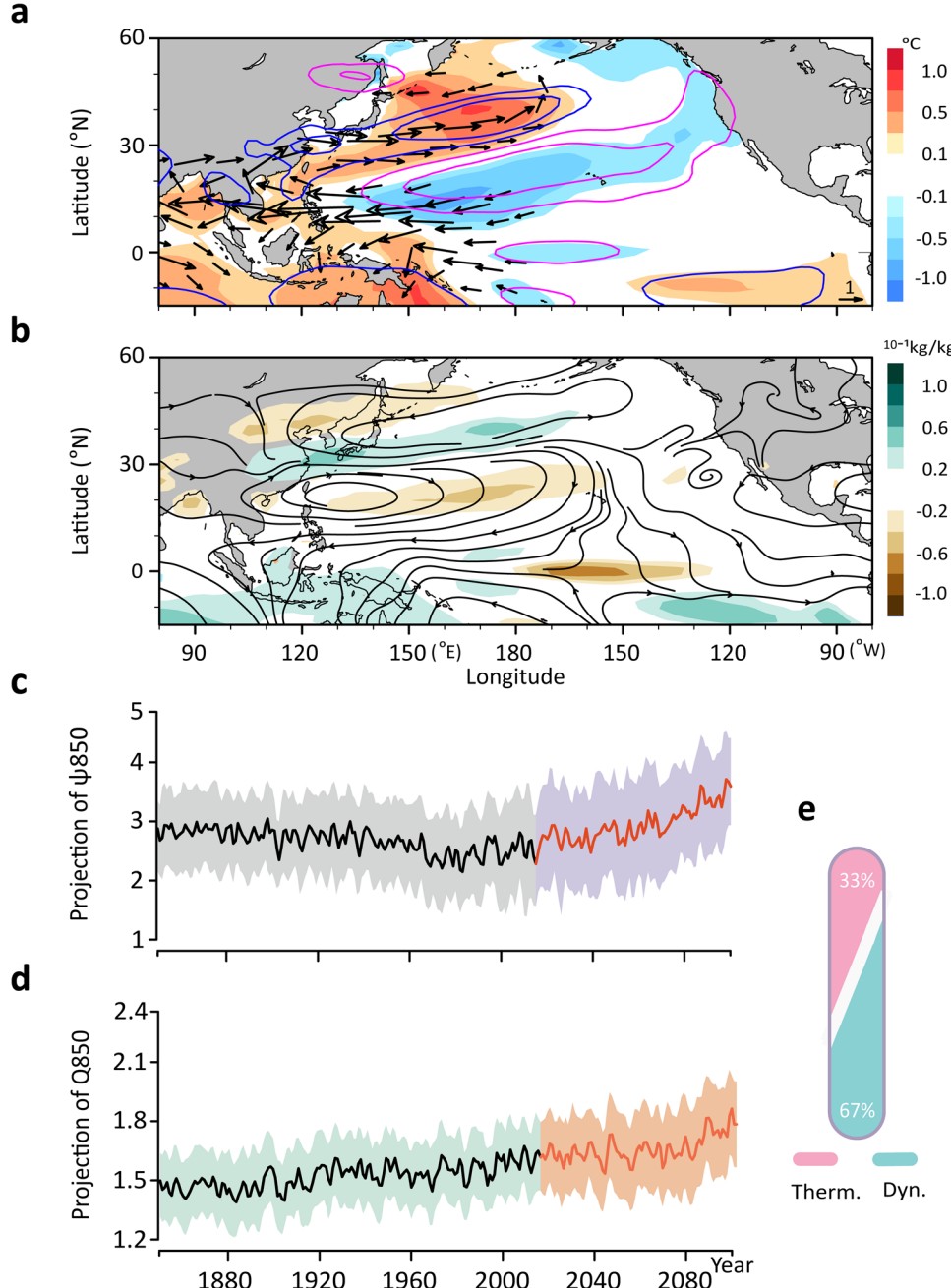

**Fig. 4 | The dynamic patterns associated with the first neutral mode (NM1).**
**a** The NM1-associated patterns in sea surface temperature (shading), 850 hPa wind (vectors, ms⁻¹), and 500 hPa geopotential height (contours, gpm). **b** The NM1-associated patterns in $Q850$ (shading) and streamline (arrowed lines). See Methods for the calculation of the NM1-associated patterns. **c** The time series of the projection of $\psi850$ onto the NM1-associated pattern. The shading indicates the two std of the internal variability of the grand ensemble. **d** Same as **c** but for the time series of the projection of $Q850$ onto the NM1-associated pattern. **e** The fractional contributions to the precipitation trend during the projection period from the dynamic (light blue) and thermodynamic (pink) components in the CESM-SMILEs.

that the magnitude of the conventional projections is much weaker than the corresponding NM-based projections. We interpret the weak magnitude in the former as the result of the cancellation across different CMIP6 models with biases in a somewhat random fashion over the Asia-Pacific sector. Indeed, if one treated the models as perfect and the grand ensemble mean of the CESM-SMILEs as the "true" response, the NM-based projection would asymptote to the grand ensemble mean with increasing number of NMs included (Fig. S6). However, if one treated one ensemble of the CESM-SMILEs as the "true" climate and other ensembles as "models" to capture the "true" climate change signal, the NM-based projections would lose the monotonicity in the skill increase with increasing number of NMs

(Fig. S7, blue dashed line). Compared to the case where all the models are treated as perfect (i.e., blue line), this illustrates the greater uncertainties and the lack of consensus in the higher-order modes among climate models. In addition, a model's uncertainty as measured by the model's spread-to-signal ratio (MSSR, red line, calculated as the ratio of the variance of the inter-model spread to the variance of the grand ensemble mean) can increase sharply as more NMs are used in the projection. Thus, there is a trade-off in the NM-based projection between the number of modes considered and the risk of contamination by the uncertain modes. Therefore, only a smaller number of NMs may be selected for the purpose of robust climate change projection.

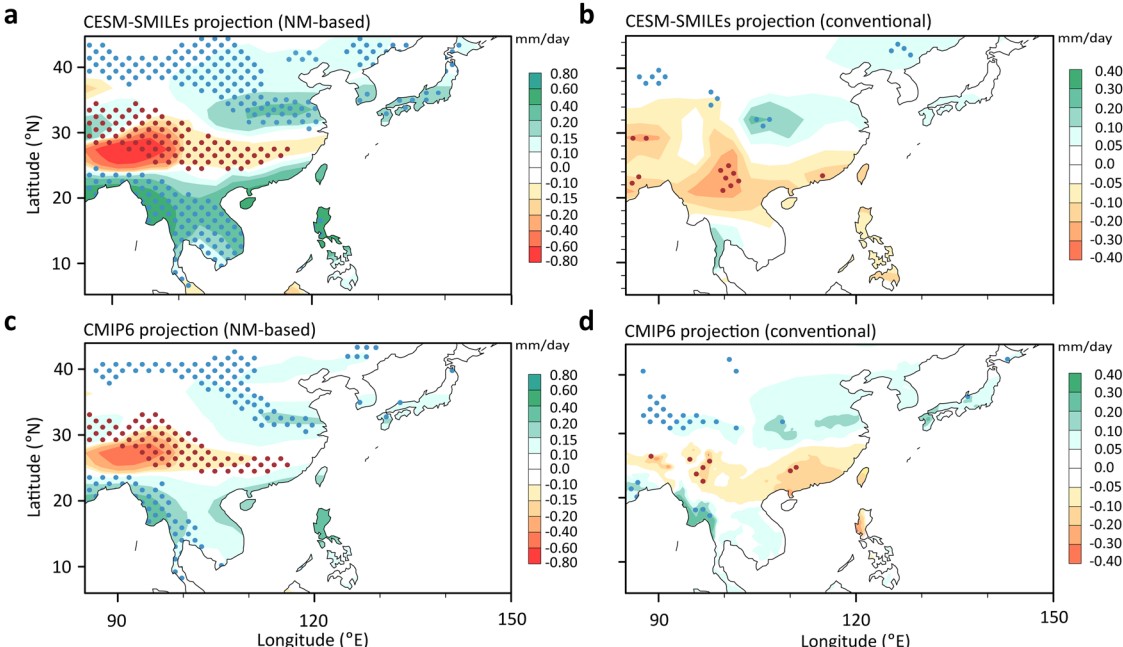

**Fig. 5 | Neutral mode (NM)-based versus conventional precipitation projection. a** Ensemble mean change of the June–July–Augst (JJA) precipitation from the synthetic ensemble based on 25 combinations of the projected changes in NM1 and NM2 (shading, mm day⁻¹) and the corresponding across-member consensus indicated by the >90% agreement on the sign of the local rainfall change by the end of the 21st century (dotted areas). **b** The projection of the JJA precipitation using the multi-ensemble mean of CESM-SMILEs. **c** Similar to **a**, but for the CMIP6 simulations projected onto the NM1 and NM2. **d** The projection of JJA precipitation based on simple multi-model ensemble mean of CMIP6.

Finally, we attempt to extend the NM-based projection to the higher order modes for CESM-SMILEs. For those modes that satisfy the "first year of ensemble mean out" condition within the 21st century with the same sign across the models, and whose inclusion does not push the averaged model spread-to-signal ratio (MSSR) to be greater than one, we consider them in the multi-NM-based projection and the result is reported in Fig. S8. See Methods for the exact procedure of the multi-NM-based projection. It turns out that considering six more NMs does not fundamentally change the pattern and magnitude of the projected precipitation change.

## Discussion

A consensus has been reached on the component of future regional rainfall change over East Asia that is projected onto the leading NMs, with the NM-based projection pattern in sharp contrast to what one would obtain from the conventional multi-model ensemble average. Given that NM is the least damped and hence the most excitable mode intrinsic to the climate system, this NM-based consensus can be interpreted as the excitation (resonance) of the internal modes by (to) the external climate forcings. Given the fact that the two leading modes find their counterparts in the observations, the robust projections onto these modes may represent something more trustworthy than the conventional, point-wise ensemble projection. To the extent that the grand ensemble spread represents the true uncertainty of the internal monsoon rainfall variability, and that these leading NMs may dominate the total response, this regional pattern of summer precipitation change shown in Fig. 5a would be something to expect to emerge before the end of the century. An immediate next question is what are the anthropogenic forcings behind this consensus. Although there is little doubt that increasing greenhouse gas forcing plays a dominant role in the 21st century trend, we cannot rule out a possible contribution from the changing aerosol emissions. Indeed, we find a consistent negative trend in the NM2 index during the 20th century in the models

participating in the Detection and Attribution Model Intercomparison Project[48] (DAMIP). The upward NM2 trend discussed earlier may reflect the projected reduction in pollutant emissions[49,50]. Projections of the forced changes in the leading NMs from each of the climate-forcing agents, and the roles they may have played in the past century, belong to topics for future investigation.

An obvious caveat of the current study is that the NMs used for projection here are those of a single climate model; a more judicious approach would be for each model to project the simulated precipitation onto its own NMs. Therefore, it is imperative to identify the linear response function and the associated NMs for each model. As such, an inter-model mode comparison project similar to that proposed by ref. 51 is needed to further solidify the robustness of the projection. If the NMs to be identified for multiple models display agreement, potential emergent constraints[52] would be more readily established from the NM perspective, with a hope of narrowing the spread of the projected changes in the leading modes. It is also important to note the difference between robustness and accuracy of the regional climate projection: the NM-based approach can bring out the robust aspect of the regional response, but it does not necessarily lead to more accurate regional climate projection. The latter will be predicated on building agreements on higher and higher orders of the NMs across climate models.

Equally important is the transferability of the future NMs projections from climate models to reality, the basis of which is mainly based on the spatial similarity between the NMs and the observed EOFs. This transferability can be further strengthened by the agreement on the NM patterns across different climate models, a consensus that has yet to be reached, and by projections to each model's own NMs. While it is unlikely that the NM-based projection will fully capture the true climate change response–an impossible task–it provides a feasible pathway to approach the true climate change signal as climate models improve and model uncertainty is gradually reduced.

## Methods

### Linear response function and neutral modes from Green's function experiments

The detailed procedure for constructing the LRF for the JJA global precipitation using Green's function forcing perturbation experiments is reported in Lu et al.[30]. Here only a brief recapitulation is provided.

While several methods have been used in the climate community to estimate the LRF of the climate system, such as the linear inverse model (LIM[53]) and the fluctuation-dissipation theorem[54], the Green's function forcing perturbation approach has proven to be more robust and applicable to more general systems, such as the Earth's climate[29]. The model used for our Green's function experiments is the CAM5 coupled to a motionless mixed-layer ocean; the experiment set consists of a 150-year control run as reference and 99 pairs of q-flux perturbation runs. Each pair consists of two experiments with q-flux anomalies of equal magnitude but opposite sign from a rectangular patch representing a geographic location of the global ocean. The 99 q-flux anomaly patches are tiled in such a way so that their sum is equivalent to a uniform flux of 12 Wm$^{-2}$ over the global open ocean. The paired forcing perturbations allow us to isolate the linear response and the corresponding linear response function, as the climate system represented by models can have rather sizable nonlinear dependence on the sign and magnitude of the forcing if it is not small enough[51,55].

Assuming a common linear operator that governing the linear response of the JJA precipitation to each of the perturbation cases, there is a systematic linear relationship between the forcing and response as follows

$$\mathscr{L}\delta\boldsymbol{P} \approx \boldsymbol{F}, \qquad (3)$$

where $\delta\boldsymbol{P}$ comprises 99 columns of vectors ($\delta\boldsymbol{p}_k$), each representing the linear JJA precipitation response to the corresponding q-flux forcing (the linear response $\delta\boldsymbol{p}_k$ for each patch $k$ is calculated as the half of the difference of the response to the positive forcing perturbation minus that to the negative forcing perturbation), $\boldsymbol{F}$ is a 99 by 99 diagonal matrix whose diagonals are the area-integrated q-fluxes of the patches. Making use of the diagonality of $\boldsymbol{F}$, it is easy to compute the Green's function matrix $\boldsymbol{G}$ as $\delta\boldsymbol{P}\boldsymbol{F}^{-1}$, with its $k$th columns computed as

$$\boldsymbol{g}_k = \frac{\delta\boldsymbol{p}_k}{F_k}, \qquad (4)$$

where $F_k$ is the area-integrated q-flux within patch $k$. Finally, the Green's function-based LRF $\mathscr{L}$ can then be estimated as the pseudo-inverse of the Green's function matrix: $\mathscr{L} \approx \boldsymbol{G}^{-1}$.

Applying singular value decomposition (SVD) to matrix $\mathscr{L}$, i.e.,

$$\mathscr{L} = \boldsymbol{U}\boldsymbol{\Sigma}\boldsymbol{V}^T, \qquad (5)$$

one can then extract the non-dimensional neutral modes of the forcing-response system represented by the right vectors stored in $\boldsymbol{V}$ and their corresponding optimal forcing patterns represented by the left vectors stored in $\boldsymbol{U}$. The singular values ($s_i$) are all negative and are contained in the diagonal matrix $\Sigma$ in an ascending order, with the first singular value being the smallest (in terms of absolute value), representing minimum dissipation and maximum excitability.

We test the separability of the singular values of the two leading NMs using Jackknife resampling, which involves a leave-one-out strategy in the estimation of the parameters (in our case, the two smallest singular values) in a data set of $N$ records. For the 99 forcing cases, we can repeat the calculation of the LRF and its two smallest singular values 99 times, leaving out one forcing case each time. We then use a paired-sample one-tailed $t$ test to test the null hypothesis

that the difference between the 2nd smallest and the smallest singular values is zero, against the alternative that the difference is nonzero. The $t$ test rejects the null hypothesis at the 5% significance level, indicating that the two smallest singular values and their corresponding singular modes are well separated. The same test is also performed for the separability between the 2nd smallest and the 3rd smallest singular values, and the result also rejects the null hypothesis.

The associated patterns with a mode of interest in other variables ($X$), such as sea surface temperature, geopotential height, etc., can be obtained by the following procedure. From Eq. (5), since the ith precipitation neutral mode (say $\boldsymbol{v}_n$) can be thought of as the response to the corresponding rescaled left vector, i.e., $\boldsymbol{u}_n s_n$, the associated patterns of other variables can also be thought of as being driven by this rescaled optimal forcing. Since the Green's function experiments have already produced the response of $X$ to each of the patches, we can make use of them and stack the q-flux normalized response ($\frac{\delta x_k}{F_k}$) to form a response matrix $\delta\boldsymbol{X}$. The associated dimensional pattern in the variable $X$ with mode $n$ can then simply be estimated as

$$\delta\boldsymbol{x}_n = \delta\boldsymbol{X}\,\boldsymbol{u}_n s_n \qquad (6)$$

### Partitioning of uncertainty

We first define the internal variability uncertainty for the variable $x$ in the JJA season of each year $t$ for each individual model $m$ as

$$I_{m,t} \equiv \mathrm{var}\left(x_{m,t,i} - \bar{x}^m{}_t\right),$$

and the model uncertainty as

$$M_t \equiv \mathrm{var}\left(\bar{x}^m{}_t - \bar{\bar{x}}_t\right),$$

respectively, where the subscript $i$ indicates the ensemble member, $(\bar{\,})^m$ the ensemble mean for model $m$, $(\bar{\,})$ the multi-member, multi-model ensemble mean, which hereafter is referred to as the grand ensemble mean. The internal variability uncertainty is computed as the multi-model ensemble mean of the internal variability uncertainty of each model: $\bar{I}_t$; thus the total uncertainty at a given year $t$ can be estimated as the sum of the multi-model ensemble averaged internal variability uncertainty and the model uncertainty, i.e.,

$$T_t = \bar{\bar{I}}_t + M_t$$

The uncertainty partitioning for the std of the NMs can be computed similarly but for the std ($\sigma$) of the daily NM index variability for a given summer. Defining a reference "historical" value $x_o$ as the grand ensemble mean of $x$ averaged over the period from 1850–2010, the "future" forced grand ensemble response in this study is computed as $\bar{\bar{x}}_t - x_o$.

### Criteria for choosing NMs for precipitation projection

For a NM to be included in the NM-based projection, the sign of the ensemble mean projection on that mode must agree among all models of the CESM-SMILEs. In addition, for each of the selected NMs, its projected trend during the 21st century must exceed "the first year ensemble mean out" criterion. Finally, to be considered in the NM-based projection, the inclusion of the additional NM must not increase the averaged model spread-to-signal ratio (MSSR), defined as follows, to greater than one.

$$\frac{1}{\mathscr{N}}\sum_n N_n/S_n$$

where $N_n$ indicates the variance of the across-ensemble spread of the projection to the $n$th mode due to model uncertainty, $S_n$ is the variance

**Table 1 | List of model simulations from LENS2 and SMILEs**

| Modeling center | Model version | Resolution | Years | No. of members | Forcing | References |
|---|---|---|---|---|---|---|
| CSIRO | MK3.6 | ~1.9° × 1.9° | 1850–2100 | 30 | historical, rcp85 | Jeffrey et al.[57] |
| GFDL | CM3 | 2.0° × 2.5° | 1920–2100 | 20 | historical, rcp85 | Sun et al.[58] |
| SMHI/KNMI | EC-EARTH | ~1.1° × 1.1° | 1860–2100 | 16 | historical, rcp85 | Hazeleger et al.[59] |
| NCAR | CESM2 | ~1.0° × 1.0° | 1850–2100 | 100 | historical, ssp585 | Rodgers et al.[37] |
| CCCma | CanESM5 | ~2.8° × 2.8° | 1850–2100 | 25 | historical, rcp85 | Swart et al.[60] |

For the CMIP6 data sets, we use monthly precipitation for both the historical (1962–2005) and the future SSP5.85 scenario for the future projection (2056–2099) of the Asian monsoon rainfall. Only the first member of each model is used. Data from the following models are analyzed: ACCESS-CM2, ACCESS-ESM1-5, AWI-CM-1-1-MR, BCC-CSM2-MR, CAMS-CSM1-0, CanESM5, CESM2, CESM2-WACCM, CIESM, CMCC-CM2-SR5, CNRM-CM6-1, CNRM-CM6-1-HR, CNRM-ESM2-1, EC-Earth3, EC-Earth3-Veg, FGOALS-f3-L, FGOALS-g3, FIO-ESM-2-0, GFDL-ESM4, GISS-E2-1-G, HadGEM3-GC31-LL, HadGEM3-GC31-MM, IITM-ESM, INM-CM4-8, INM-CM5-0, IPSL-CM6A-LR, KACE-1-0-G, MCM-UA-1-0, MIROC6, MIROC-ES2L, MPI-ESM1-2-HR, MPI-ESM1-2-LR, MRI-ESM2-0, NESM3, TaiESM1, UKESM1-0-LL. The CMIP6 data can be downloaded from: https://esgf-node.llnl.gov/search/cmip6/.

of the multi-model mean projection onto the $n$th mode, and $\mathcal{N}$ is the number of the total NMs selected. The projection is performed sequentially from low to high NMs. The criteria allow us to select up to the 10th NM, excluding the 4th and 8th modes (see the result of the projecting in Fig. S8).

### Data sets

APHRODITE, a daily gridded precipitation data set for Asia spanning from 1951 through 2007, is obtained from http://www.chikyu.ac.jp/precip/ and used for the estimate of the observed EOF of the Asian summer monsoon precipitation[56].

The CESM2 Large Ensemble (CESM-LENS2) consists of 100 members at 1-degree spatial resolution covering the period 1850–2100 under the CMIP6 historical and SSP370 future radiative forcing scenarios. More details about CESM-LENS2 can be found at https://www.cesm.ucar.edu/projects/community-projects/LENS2/. For the ToE estimation and the uncertainty attribution analysis to be compared with the SMILEs simulations, the linear trends in CESM-LENS2 are all scaled up by a factor of 1.3 during the projection period (2011–2100). The projection period starts from the year when the GHG forcing was already ~2 Wm$^{-2}$ compared to the pre-industrial level. Therefore, the rescaling factor is calculated as: $(8.5 - 2)/(7 - 2) = 1.3$. With the rescaling, we can form a grand ensemble by combining LENS2 and SMILEs under a presumably same GHG forcing.

SMILEs data are archived in the Multi-Model Large Ensemble Archive (MMLEA; Table 1) at the National Center for Atmospheric Research[38]. All SMILEs simulations used here are forced with the standard CMIP5 "historical" and Representative Concentration Pathway 8.5 (RCP8.5) forcing scenarios. The SMILEs models have horizontal resolutions ranging from ~1° to ~2.5° and the ensemble size for each model varies from 16 to 30. Here we only select four large ensembles (CanESM5, MK3.6, CM3, and EC-EARTH) that provide daily precipitation data on the publicly available SMILEs website: http://www.cesm.ucar.edu/projects/community-projects/MMLEA/. All the large ensemble data used in this study are listed in Table 1.

### Data availability

In addition to the publicly available data sets mentioned in the Methods section, the Green's function data sets used to compute the summer precipitation neutral modes have been published on the publicly accessible data portal Zenodo at https://doi.org/10.5281/zenodo.4588073.

### Code availability

The code central to the main conclusions of this study will be released upon the acceptance of the manuscript on the data portal: https://zenodo.org/record/7936696#.ZGH3WXZBwuV.

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

## Acknowledgements

This work is supported by the National Key Research and Development Program of China (2020YFA0608903) and the Office of Science, U.S. Department of Energy Biological and Environmental Research as part of the Regional and Global Model Analysis program area. The Pacific Northwest National Laboratory (PNNL) is operated for DOE by Battelle Memorial Institute under contract DE-AC05-76RLO1830. D.X. and Y.Z. is jointly supported by the National Science Foundation of China (42105051, 42105166 and 42075020). F.S. is supported by the National Natural Science Foundation of China (Grant 42175029). We also acknowledge the US CLIVAR Working Group on Large Ensembles for making the SMILEs data sets publicly available and the World Climate Research Program's Working Group on Coupled Modeling, which is responsible for CMIP6.

## Author contributions

D.X. and J.L. came up with the original idea of the dynamical mode-based approach for regional climate projection, inspired by a conversation with H.T., and were also responsible for the correspondence during the review process. D.X. also performed all the analyses of the study and wrote the first draft of the manuscript. J.L. and L.R.L. contributed to drafting and revising the manuscript. H.T., F.S., T.Z., and Y.Z. contributed to the interpretation of the results as experts in monsoon dynamics and climate projections; they also made suggestions that helped improve the presentation of the work.

## Competing interests

The authors declare no competing interests.
