## [Peer Review File · Nature Communications]

Robust projection of East Asian summer monsoon rainfall through the lens of the dynamical modesREVIEWER COMMENTS

Reviewer #1 (Remarks to the Author):

Robust Projections... Xue et al. submitted to Ncomms

This is an interesting paper — but I am still not sure of what to make of its conclusions. I have a few suggestions and a bunch of questions.

Let me first summarize what I think the story of the paper is, to make sure that I do get it and that my comments are indeed relevant. The idea here is to identify (from a single GCM run in a slab ocean configuration) the least damped modes of precipitation response to surface (ocean) heat fluxes and to assess how the precipitation anomalies linked to those modes emerge as forced signal. They then proceed to show that the (future) projection of the (pattern) projection is “robust”, both in the sense that it emerges earlier and that its sign is in agreement across models, leading to a map of “NM-based rainfall projection” that looks like a known dynamical pattern as well with good model agreement everywhere where the anomalies are substantial (away from the zero line). In passing, the authors also provide a decomposition of the NM-based projection in dynamic and thermodynamic components.

What I cannot wrap my brain around is: what am I supposed to do with this projection? If I am interested in regional rainfall change, should I look at this map, instead of the CMIP6 multi-model mean? Within the logic of this work, why should one trust the NM-based projections more? Is the disagreement that mars the traditional multi-model mean an indication of noise, or is it something else that ruins an otherwise robust projection? I just don't understand in what way I should interpret the difference between the two projection methods.

Figure 3 is a good starting point to discuss this question, as it shows both the NM-based and standard projection for CMIP6. At the very least, the authors should do the same for CESM-LE, the other SMILEs models and the CESM-SMILEs grand ensemble. This would provide a measure of what one gains from using the NM method in some sort of “perfect model” set up, where we know the projection very well (because of the large ensemble, which eliminates scenario and model uncertainty by design and beats down internal variability uncertainty to a minimum).

The other elephant in the room is the fact that the NMs are extracted from just one model. Would the result change if I were to use a different model? Especially if the implication of this study is that one should look at the NM-based projections, the sensitivity to the choice of the base vectors need to be fully investigated. The discussion section admits the importance of these caveats, but I think that more should be done to address them.

For example, even without the ability of calculating the Neutral Modes for other CMIP models, one could at least report on how different the EOFs are across the ensemble. Given that there is a close correspondence between EOFs and NMs in CESM, it would provide the reader to some context for how robust this method is to the choice of the NMs. Much better, of course, would be to duplicate the method with a different model.

In other areas of the world (e.g in the African monsoon), the fingerprint of the forced response changes from historical to future climates. How do you know that the changes in the projection onto the NM will remain relevant? Can you show that the NM remains unchanged across climate states? (Again, at the very least one can show that for the EOFs).

Finally, I am confused about the method for separating thermodynamic and dynamic components of the change. First of all: you are again using only one model, and that should be said upfront (L219). This is especially important, given that the trend shown in Figure 2a is somewhat anomalous compared to the others. In fact, the time series for the rainfall NM1 actually looks stationary after 2060, but the stream function seems to be picking up after that. Second, how do you calculate the percentages? Are you taking a linear trend in NM1 over the century (call it P%)

and one for the 850mb stream function(call it S%) and say that the dynamic component is S/P? Is this estimate robust to the choice of dynamical variable? Is this consistent with estimates from moisture budget analysis? From experiments separating uniform and pattern warming? I feel that this needs more discussion.

The paper is well organized and reads well, aside from some scattered English usage mistakes. These are small things ,such as when to use or drop articles, but they add up. I suggest getting some editing help.

Details

L52: what do you mean by "unique dynamic and thermodynamic constituents and their complex interplay"? How is this an explanation for why projections are uncertain? What is unique about this situation?

L65: it would be a good idea to start with a more explicit definition of the neutral modes. I find a comment such as "often preferentially excited" to be confusing. (You have better explanations right after this (L68) and in the next section (L91), so you just need to shuffle things around)

L73 I admit to not having read references 20 and 21. Still, the idea that the precipitation response determined this way is less dependent on the physics of a model strikes me as unlikely — as well as ill-defined: I assume you are not saying it depends on the dynamical core!

L119: one way around this, at least partially, is to do this analysis in terms of change per degree of global warming and inflate the uncertainty at the end by rescaling including the scenario uncertainty.

L121: You should be explicit here about the fact that the NM have been calculated only for one model, but used for projections by all models. You discuss the implications of this later, but it would be good to spell it out earlier as well.

L135: I am not sure what to make of this... You seem to imply that you are doing exceptionally well, but I worry that the significance is that our models are not properly capturing the full spectrum of variability in the region.

L140: The variance about the mean in Figure 2 is very different across the models, even though you are plotting it in terms of standard deviation and not actual spread, which should make the size of the ensemble unimportant. What do you believe is the reason for this?

L199: Can this have something to do with the fact that NM1 from CAM5 is not optimal for this model? This should be discussed.

L206: To the casual reader, labeling panel (e) as JJA mean (or (f) as JJA std) will lead to confusion. I am also not fond of this labeling in the other part of the figure (mostly because projection here means something quite different from "climate projection", but one needs to think it through). How about you substitute NM1 for JJA?

L228: See above.

L274 (Figure 3a): typo in the title: CESM-SMILEs.

L331: "more general" than what?

Reviewer #2 (Remarks to the Author):

The authors present the future changes in the East Asian summer monsoon by projecting the CMIP6 model forecasts onto the two leading dynamical modes of internal variability. Even though there is a considerable uncertainty in the CMIP6 projected changes to the East Asian summer monsoon, the authors find that there is a remarkable consensus among the models when only the

two leading neutral modes (NMs) are used. They find the precipitation trend is dominated by dynamical changes, with variability increasing with time and also show the time of emergence of these changes. These results are novel and interesting, but there is still some work which needs to be done before this paper can be considered for publication.

Major concerns:

1. As shown in Fig5, even though the NMs-based projections capture the broad features of precipitation changes, its spatial pattern (especially for the region of drying) differs from the CMIP6 MMM projections. What are the reasons for this? Why are the CMIP6 conventional projections about an order of magnitude smaller than the NM-based projections?
2. On a related note, given this mismatch between CMIP6 forecasts and NM-based projections, what are the justifications for treating the leading modes as a proxy for the total precipitation changes. From the practical viewpoint, wouldn't the total CMIP6 projections be more useful than the leading modes.
3. The neutral modes are obtained by perturbing the ocean q-flux. But wouldn't this have a difficulty in capturing the faster acting influences of direct CO₂ forcing not related with SST warming. Regional precipitation patterns are known to be sensitive to these direct CO₂ effects and land warming.
4. Is noise the major factor that makes CMIP projections of East Asian summer monsoon to be non-robust as compared to the top two neutral modes?
5. Given that NM1 and NM2 explain only about 50% of the total variance in precipitation trend, would it be better to include the higher modes. Can adding these additional modes make the sum to be non-robust even though the leading neutral modes are robust.
6. I am not convinced that the model consensus of NMs-based projections can be treated as the consensus for the future projections of East Asian summer monsoon. As pointed in line 286, the results/inferences are constrained by the assumption that the leading neutral modes dominate the total response. More work needs to be done to demonstrate this.

Minor comments :

Fig1 and in other figs too, the colors on the red spectra (1b and 1c) are confusing. It's darker in the middle and is lighter on either side, and it's somewhat difficult to distinguish the red shades. Perhaps use a different colorbar like in Fig 3a for example.

Fig1a : define what the gray land regions represent. Also it might be worth mentioning that the colorbars are in log scale.

It might be a good idea to show a similar analysis as in Fig2 for the total precipitation projections, perhaps in supporting information.

Typos:

L200: NM2 index

L325: Ref number for Lu et al is 22.

First of all, we'd like to acknowledge the perceptive comments of the two reviewers, which help improve the quality and readability of the manuscript substantially and help us contextualize our study in a broader spectrum of climate change and climate projection research. During the revision, we realize the importance of underscoring the close lineage of our approach to the celebrated '*fingerprinting*' approach pioneered by K Hasselmann. In essence, our approach is to make climate change projection or detection using the neutral modes (NMs) as the fingerprints—a means to overcome the challenge of the curse of dimensionality. During the revision we now make the connection of our NM-based approach to the idea of the fingerprinting directly (see L71-88). With the bridge so built, many of the comments raised by the reviewers can be naturally addressed by leveraging the literature on the fingerprinting approach. In the following, we try to address the review comments to the best of our capability. Please note that the line numbers referenced herein are those of the tracked version of the revised manuscript.

Reviewer #1

This is an interesting paper — but I am still not sure of what to make of its conclusions. I have a few suggestions and a bunch of questions.

Let me first summarize what I think the story of the paper is, to make sure that I do get it and that my comments are indeed relevant. The idea here is to identify (from a single GCM run in a slab ocean configuration) the least damped modes of precipitation response to to surface (ocean) heat fluxes and to assess how the precipitation anomalies linked to those modes emerge as forced signal. They then proceed to show that the (future) projection of the (pattern) projection is “robust”, both in the sense that it emerges earlier and that its sign is in agreement across models, leading to a map of “NM-based rainfall projection” that looks like a known dynamical pattern as well with good model agreement everywhere where the anomalies are substantial (away from the zero line). In passing, the authors also provide a decomposition of the NM-based projection in dynamic and thermodynamic components.

What I cannot wrap my brain around is: what am I supposed to do with this projection? If I am interested in regional rainfall change, should I look at this map, instead of the CMIP6 multi-model mean? Within the logic of this work, why should one trust the NM-based projections more? Is the disagreement that mars the traditional multi-model mean an indication of noise, or is it something else that ruins an otherwise robust projection? I just don't understand in what way I should interpret the difference between the two projection methods.

Your questions touch the core of the problem of climate projection. To address your question, (since we can't say better than Hasselmann) we like to quote/paraphrase the seminal work of K Hasselmann (1979): “The determination of the mean atmospheric response in the presence of noise/internal variability presents a basic signal detection problem. Because of the limited length of data record/model simulations, the signal-to-noise problem can then become one of the severest limitations in the study of the atmospheric response to external forcing ... Indeed,

the signal-to-noise analysis has largely been formulated in previous studies in terms of individual gridpoint statistics, rather than the pattern response ... the question of whether or not the response pattern, as a whole or in part, is statistically significant clearly cannot be resolved by such an approach ... if *a priori* hypotheses regarding the general structure of the expected response can be formulated, pattern filtering can be applied to improve greatly the signal-to-noise ratio ... The consistent application of pattern filtering techniques may therefore be expected to enhance considerably the effectiveness of GCM experiments in studies of the atmospheric response to external forcing.”

Our approach here is essentially a fingerprinting approach (or pattern filtering, in other words) for monsoon rainfall projection. During the revision, we realize that the work of Hasselmann and similar works by others set a nice backdrop for our study here and we now make an explicit connection to the fingerprinting idea. Given this context, the main new idea of our study is to introduce the use of neutral modes (NMs), which have dynamical underpinning, for pattern filtering/fingerprinting of climate projections while fingerprinting was performed mainly using empirical orthogonal functions in previous studies in the context of climate change detection (see our revised Introduction paragraph).

Meantime, we must acknowledge that this is the first attempt on regional precipitation projection in reduced space/dimension. The result here (as reported in Figure 5) is not final, but we believe this is a good starting point to make better use of the CMIP simulations despite their caveats and biases. Our results also suggest the NM-based approach to be more promising, compared to the conventional pointwise approach in providing more robust regional projections. Interpreted using the filtering/fingerprinting concept, “the disagreement that mars the traditional multi-model mean” is indeed an indication of noise at grid scale. Filtering through patterns suppresses the noise to increase the signal-to-noise ratio, providing more robust projections of future changes. In short, the NM-based projection should be more trustworthy than the pointwise one, although the projections we made here remain to be completed, for example, by extending to more dynamical modes that are robust, such as regional responses to orographic forcing.

Figure 3 is a good starting point to discuss this question, as it shows both the NM-based and standard projection for CMIP6. At the very least, the authors should do the same for CESM-LE, the other SMILEs models and the CESM-SMILEs grand ensemble. This would provide a measure of what one gains from using the NM method in some sort of “perfect model” set up, where we know the projection very well (because of the large ensemble, which eliminates scenario and model uncertainty by design and beats down internal variability uncertainty to a minimum).

We assume this comment is referring to Figure 5 instead of Figure 3, as Figure 5 shows the NM-based projection in comparison with the standard projection for CMIP6. In Figure 5, we already showed a comparison of the NM-based projection and the conventional based projection for CESM-SMILEs (i.e., the grand ensemble of 5 models including CESM2, CanESM5, CM3, EC_Earth, MK36). Do you want us to compare the NM-based projection to the standard projection for

each of the ensemble of CESM-SMILEs? If we get your suggestion correctly, that was what we examined first in this paper (though not shown).

In the revised Figure 5, we now compare the standard projections with the NM-based projections for both CMIP6 and CESM-SMILEs. We also discussed the result in the text: “In view of the fact that the leading two modes find their counterparts in the observations, the robust projections onto these modes may represent something more trustworthy than the simple multi-model ensemble projection without discretion.”

The “perfect model” approach is not our intent here. As we stated in Introduction, models suffer considerable biases over Asian monsoon region, we cannot treat the ensemble mean of a climate model (say, CESM-LE) as the truth. Our study tackles the signal-to-noise issue and model biases simultaneously.

The other elephant in the room is the fact that the NMs are extracted from just one model. Would the result change if I were to use a different model? Especially if the implication of this study is that one should look at the NM-based projections, the sensitivity to the choice of the base vectors need to be fully investigated. The discussion section admits the importance of these caveats, but I think that more should be done to address them. For example, even without the ability of calculating the Neutral Modes for other CMIP models, one could at least report on how different the EOFs are across the ensemble. Given that there is a close correspondence between EOFs and NMs in CESM, it would provide the reader to some context for how robust this method is to the choice of the NMs. Much better, of course, would be to duplicate the method with a different model.

This is a good point you bring up. The key justification for using the NMs from CAM5-SOM is their correspondences with the observed EOF patterns over the Asia-Pacific domain. This serves as a mutual validation: the NMs are likely reflecting the true modes; and the EOFs from the observation are representative of the dynamical modes of the monsoon system. This is the key foundation for the whole study. If the NMs from a different model differed substantially from the observed EOFs, they shouldn't be used for climate projection purpose, because this would imply severe bias of the model.

We did investigate how different the EOFs are across the ensemble members for each model and across different models. There is a screening process to decide which models to include and how many modes to keep in the NM-based projection. In the original version, because of the space limit, we did not include such discussion. Now we have included a justification for the choice of EOFs/NMs for the NM-based projections in Section **Projection from a Synthetic Ensemble**.

For your information, we include the pattern correlations between the EOF1 (EOF2) computed from two random ensemble member of the CESM large ensemble in Table R1 below (showing that the leading EOF patterns are robust across different members of a same model); and the pattern correlations between the EOF1 (EOF2) computed from each pair of two different

models (or a model and the observation) from CESM-SMILEs in Table R2. For the values in Table R2, we also perform non-parametric statistical tests for the significance of the correlations. As you can see, there is systematic significant correlations for the EOFs from the CESM and SMILEs models. As a reminder, the EOF1 and EOF2 from CESM correspond spatially well with NM1 and NM2, respectively. This gives us some confidence to use the ensemble data from SMILEs for the NM-based projection. On the other hand, no systematic pattern correlation can be established across CESM-SMILEs for EOF3 and EOF4.

Table R1 Pattern correlations of EOF1 and EOF2 across random members in CESMILE2.

Mode2 Mode1	Ens.18	Ens.01	Ens.05	Ens.11	Ens.19	Ens.03
Ens.18	1	0.96	0.95	0.91	0.97	0.89
Ens.01	0.99	1	0.96	0.92	0.97	0.86
Ens.05	0.96	0.96	1	0.94	0.94	0.88
Ens.11	0.98	0.96	0.95	1	0.94	0.91
Ens.19	0.96	0.96	0.95	0.95	1	0.87
Ens.03	0.96	0.97	0.94	0.96	0.95	1

Table R2 Pattern correlations of EOF1 and EOF2 across different models (or between a model and the observation)

Mode2 Mode1	CanESM5	CM3	EC_EARTH	MK36	CESM2	Obs.
CanESM5	1	0.50*	0.41*	0.18	0.44*	0.32*
CM3	0.61*	1	0.29	0.27	0.66*	0.52*
EC_EARTH	0.42*	0.66*	1	0.32*	0.79*	0.81*
MK36	0.54*	0.51*	0.20	1	0.51*	0.70*
CESM2	0.40*	0.69*	0.27	0.30	1	0.48*
Obs.	0.34*	0.49*	0.43*	0.56*	0.60*	1

Last, we agree with you it is imperative to extend our NM exercise here to other models; Green’s function experiments inter-comparison project may be in order.

In other areas of the world (e.g in the African monsoon), the fingerprint of the forced response changes from historical to future climates. How do you know that the changes in the projection onto the NM will remain relevant? Can you show that the NM remains unchanged across climate states? (Again, at the very least one can show that for the EOFs).

The basis for the NM-based projection is the linear perspective embodied by the following equation for equilibrium forcing-response relationship:

$$Lx = f \quad (R1)$$

L can be thought of as the Jacobian operator (a linear operator) with respect to a given background state. As such, NMs are also dependent on the background mean state. This linear view has been shown to be valid by the seminal work of Grant Branstator:

“Modes of Variability” and Climate Change, *J. Clim.*, 22, 2639-2658,
<https://doi.org/10.1175/2008JCLI2517.1>

According to Branstator, the response of the climate system to external forcing will tend to have a structure that is similar to the structure of one or a few of the system’s leading intrinsic modes of variability, which are a function of the present mean climate state. You are right that the modes do change as climate changes. But the changes of the modes should not be directly related to the forced response within the linear framework. Therefore, the change of the NMs is not of concern for our purpose here. It can be another interesting topic to explore in the future. But, before one can establish consensus on the modes of the present-day climate, one should not hasten to look at the change of the modes.

Finally, I am confused about the method for separating thermodynamic and dynamic components of the change. First of all: you are again using only one model, and that should be said upfront (L219). This is especially important, given that the trend shown in Figure 2a is somewhat anomalous compared to the others. In fact, the time series for the rainfall NM1 actually looks stationary after 2060, but the stream function seems to be picking up after that. Second, how do you calculate the percentages? Are you taking a linear trend in NM1 over the century (call it P%) and one for the 850mb stream function (call it S%) and say that the dynamic component is S/P? Is this estimate robust to the choice of dynamical variable? Is this consistent with estimates from moisture budget analysis? From experiments separating uniform and pattern warming? I feel that this needs more discussion.

First of all, there was a miscommunication between the lead author and the corresponding author on the data used for the dynamic attribution reported in our original manuscript. In fact, all the CESM-SMILEs runs are used for the attribution analysis. We now clarify this point in the revised Figure 4 and its caption.

In response to this comment, we take the mode perspective and isolate the 850hPa moisture (Q850) pattern concomitant with the leading modes (using equation (4) in Methods). Once we obtain the dimensional Q850 pattern of the NM1, we project the ensembles of the Q850 time series on to it, the resultant grand ensemble mean and the ensemble spread are plotted in the revised Figure 4 (reproduced here as Figure R1 below). But the trend of this index during the “future” projection period can only be interpreted as the “proxy thermodynamic” contribution to the trend in NM1. This “proxy thermodynamic” factor increases by only 13.8% over the projection period (compared to the 29.2% increase in the precipitation NM1 trend). However, one must keep in mind that not all the Q850 increase will be translated into increase of precipitation, as we have shown in an earlier study that both the *moisture participation ratio*

and the *hydrological cycling rate* tend to decrease with global warming (see Xue et al. 2018: “Response of the hydrological cycle in Asian monsoon systems to global warming through the lens of water vapor wave activity”, *Geophys. Res. Lett.*, 45, <https://doi.org/10.1029/2018GL078998>). Therefore, the actual *thermodynamic* contribution to the precipitation trend over the projection period should be less than $13.8\%/29.2\%=47\%$, leaving most of the trend to the *dynamical* factor. This is in keeping with the 19.6% fractional increase in the NM1-associated dynamical pattern as shown in Figure 4c (contributing 67% of the 30% increase of the precipitation NM1 index), allowing us to argue that the increase of precipitation NM1 is to a larger extent driven by dynamical factors. For your reference the revised Figure 4 is shown below as Figure R1.

Figure R1 The dynamic patterns associated with NM1. (a) The NM1-associated patterns in sea surface temperature (shading), 850 hPa wind (vectors, ms^{-1}), and 500hPa geopotential height (contours, gpm). (b) The NM1-associated patterns in 850 hPa water vapor (shading) and streamline (arrowed lines), and 500hPa geopotential height (contours, gpm). See Methods for the calculation of the NM1-associated patterns. (c) The time series of the projection of 850hPa streamfunction ψ_{850} onto the NM1-associated pattern. The shading indicates the two std of the internal variability of the grand ensemble. (d) Same as (c) but for the time series of the projection of Q_{850} onto the NM1-associated pattern. (e) The fractional contributions to the trend during the projection period from the dynamic (light blue) and thermodynamic (pink) components in CESM-SMILES.

L52: what do you mean by “unique dynamic and thermodynamic constituents and their complex interplay”? How is this an explanation for why projections are uncertain? What is unique about this situation?

We change “unique” to “specific”, by which we mean the specific land-sea-topography configuration for each monsoonal region, the vegetation coverage over the monsoonal land, and the specific ocean conditions and ocean dynamical modes of variability surrounding the monsoonal land, etc.

L65: it would be a good idea to start with a more explicit definition of the neutral modes. I find a comment such as “often preferentially excited” to be confusing. (You have better explanations right after this (L68) and in the next section (L91), so you just need to shuffle things around)

Thanks for the suggestion. This section has been completely revised. See the revised L71-88.

L73: I admit to not having read references 20 and 21. Still, the idea that the precipitation response determined this way is less dependent on the physics of a model strikes me as unlikely — as well as ill-defined: I assume you are not saying it depends on the dynamical core!

We hope the overhaul of this chapter and the response above should clarify the point for you. This point is made from the dynamical system point of view. The deterministic dynamics is different from the dynamical core in climate modeling that solves the equations of states with physics as source/sink terms; it refers to the dynamics behind the emergent patterns that can give high signal-to-noise ratio for climate change detection/projection, thus more confidence in climate projection. The dynamics is determined by both dynamical and physical processes, and we have removed the mention of unresolved physical processes in this sentence.

L119: one way around this, at least partially, is to do this analysis in terms of change per degree of global warming and inflate the uncertainty at the end by rescaling including the scenario uncertainty.

Thanks for this very good suggestion. During the estimate of ToE and uncertainty attribution

analysis, we now scale up the trends (for both the JJA mean NM indices and the std of the NMs indices) in CESM-LE during the “future” projection period by a factor of 1.3.

The projection period starts from the year (2011) when the GHG forcing was already around 2Wm^{-2} compared to the preindustrial level. Therefore, the rescaling factor is computed as: $(8.5 - 2)/(7 - 2) = 1.3$. With the rescaling, we can form a grand ensemble by combining LENS2 and SMILEs under equivalently the same GHG forcing.

See the description of the model ensembles in L182-185 for detail.

L121: You should be explicit here about the fact that the NM have been calculated only for one model, but used for projections by all models. You discuss the implications of this later, but it would be good to spell it out earlier as well.

Thanks for pointing this out. We have spelled it out explicitly here. See the revised L198-200.

L135: I am not sure what to make of this... You seem to imply that you are doing exceptionally well, but I worry that the significance is that our models are not properly capturing the full spectrum of variability in the region.

Revisiting this point, we realize that it is not a fair comparison with the variance explained by the EOFs in the observation. Here the variance of the trend pattern is a spatial variance. So, to avoid any confusion, we deleted the sentence “As a comparison, we note that the two leading EOFs of the observed JJA precipitation explain only about 30% of the variance in the same domain”.

L140: The variance about the mean in Figure 2 is very different across the models, even though you are plotting it in terms of standard deviation and not actual spread, which should make the size of the ensemble unimportant. What do you believe is the reason for this?

We can only speculate about the possible reasons. First of all, as we have found in our earlier study (Lu et al. 2021, “The leading modes of Asian summer monsoon variability as pulses of atmospheric energy flow”, *Geophys. Res. Lett.*, DOI: 10.1029/2020GL091629), water vapor and cloud can give substantial positive feedbacks to the leading neutral modes of the Asian monsoon rainfall variability. Feedbacks like this can inflate the internal variability. We also know that CESM2 has quite large a climate sensitivity compared to the median CMIP6 models. The second possible reason is simply that CESM2 has many more ensemble members than the SMILEs ensembles. As pointed out by Wood et al. (2021, Changes in precipitation variability across time scales in multiple global climate model large ensemble. *Env. Res. Lett.*, 16, <https://doi.org/10.1088/1748-9326/ac10dd>), “models with ‘small’ initial-condition ensembles systematically underestimate precipitation variability”. A third possible reason is that CESM2 is already too wet over the Asian monsoon region. The figure attached below compares the JJA climatological mean precipitation of CESM2 against GPCP data. CESM2 is overall too wet. This

can potentially provide a background for more extreme internal variability. Given the complex factors behind the large variance of CESM and the space limit, we only add a sentence in L205-207, noting that “For reasons remaining to be understood, the CESM-LE has a much larger internal variability in the NM indices than other ensembles”.

Figure R2 Comparison of the JJA mean rainfall with GPCP observation. (a) GPCP JJA rainfall climatology (mm/day); (a) CESM2 JJA rainfall climatology (mm/day); (c) Difference of CESM2 climatology minus GPCP climatology.

L199: Can this have something to do with the fact that NM1 from CAM5 is not optimal for this model? This should be discussed.

See the response to the previous points. We think this is the direct consequence of too large an internal variability. It is now referenced briefly in the revised text (L303-305):

“the stringent ToE cannot be detected for NM1 index in CESM-LE within the century due to its large internal variability and relatively weak trend (see Figure S5).”

L206: To the casual reader, labeling panel (e) as JJA mean (or (f) as JJA std) will lead to confusion. I am also not fond of this labeling in the other part of the figure (mostly because projection here means something quite different from “climate projection”, but one needs

to think it through). How about you substitute NM1 for JJA?

Revised as suggested.

L228: See above.

L274 (Figure 3a): typo in the title: CESM-SMILEs.

Amended.

L331: “more general” than what?

Revised to be “complex”.

Reviewer #2 (Remarks to the Author):

The authors present the future changes in the East Asian summer monsoon by projecting the CMIP6 model forecasts onto the two leading dynamical modes of internal variability. Even though there is a considerable uncertainty in the CMIP6 projected changes to the East Asian summer monsoon, the authors find that there is a remarkable consensus among the models when only the two leading neutral modes (NMs) are used. They find the precipitation trend is dominated by dynamical changes, with variability increasing with time and also show the time of emergence of these changes. These results are novel and interesting, but there is still some work which needs to be done before this paper can be considered for publication.

Major concerns:

1. As shown in Fig5, even though the NMs-based projections capture the broad features of precipitation changes, its spatial pattern (especially for the region of drying) differs from the CMIP6 MMM projections. What are the reasons for this? Why are the CMIP6 conventional projections about an order of magnitude smaller than the NM-based projections?

In response to this comment and a comment from reviewer #1, we now have included the conventional ensemble mean projection in Figure 5 together with the NM-based projection. As you can see, the conventional ensemble mean projections using CESM-SMILEs and CMIP6 models are more similar to each other than are the conventional vs. NM-based projections using the same data set. As you already noted, the magnitude of the conventional projection is much weaker than the NM-based projection. Here is our tentative interpretation. Within the neutral mode framework, any response to external forcing δf^+ can be decomposed as

$$\delta x \approx V \Sigma^{-1} U^T \delta f^+ = \sum_{i=1}^n \mathbf{v}_i \frac{\langle \mathbf{u}_i, \delta f^+ \rangle}{\sigma_i}, \quad (\text{R2})$$

where v_i and u_i are the column in V (right singular vectors) and U (left singular vectors), respectively. This singular mode decomposition of the forced response, ranked in terms of singular values, can be used to filter out the not-so-confident or “noisy” information in δx . For instance, the higher modes are often inaccurately represented, as they are often of smaller scales, more susceptible to the noise and uncertainties in the system. According to (R2), the total response δx can be interpreted as the summation of the modes excited by the forcing. For the conventional projection, each of the models used for projection tends to have its own bias in capturing the mode patterns with the higher modes more susceptible to the influence of noise and uncertainties. Provided these modes are biased in a somewhat random way, they will cancel each other out when being added together in the conventional projection. As a result, the magnitude of the conventional projection is very much muted due to the cancellation. For the NM-based projections, on the other hand, we use the same leading two NMs as pattern filter of the forced precipitation signal, and the precipitation data from all models are projected onto the same NM patterns (which are extracted from CAM5-SOM using Green’s function experiments). As a result, the magnitude tends to be preserved.

2. On a related note, given this mismatch between CMIP6 forecasts and NM-based projections, what are the justifications for treating the leading modes as a proxy for the total precipitation changes. From the practical viewpoint, wouldn’t the total CMIP6 projections be more useful than the leading modes.

We must clarify that we don’t treat the leading modes as a proxy for the total projected precipitation changes; and we never say so in the text. The NM-based projection represents arguably the more trustworthy portion of the climate change projection. The root problem of the conventional pointwise projection is that features at more regional scales are more dominated by the higher modes, which themselves are more uncertain as they are more susceptible to the contamination of noise. Please see the response to the first point of reviewer #1. This is the reason why there is little consensus on the regional projections using the conventional approach.

The NM-based approach is to extract the more trustworthy aspect of the projection via spatial filtering by the NM modes. Furthermore, our confidence in the leading two NMs of CESM is built upon their agreements with the statistically derived EOF patterns from the observations. As we discuss at the end of the manuscript, our work opens a new avenue for climate change projection, stratifying the projection in terms of the rank of the modes. With more and more modes identified and their consensus established, the trustworthy regional climate change projection will gradually emerge in an asymptotic manner.

3. The neutral modes are obtained by perturbing the ocean q-flux. But wouldn’t this have a difficulty in capturing the faster acting influences of direct CO2 forcing not related with SST warming. Regional precipitation patterns are known to be sensitive to these direct CO2 effects and land warming.

Thanks for the question, which touches upon the essence of the dynamical mode. In this study, we interpret the climate change response in terms of projection onto the leading intrinsic modes of the climate system. What is the system that the modes are intrinsic to? It is a slab-coupled atmospheric system; thus, the modes of this system are irrespective of the forms of the forcing, either from the direct CO₂ or from the q-flux. Plus, the scale of the response being discussed here is the time scale when both the land and the ocean mixed-layer are equilibrated; discerning the different response time scales between the land versus the ocean mixed-layer is also irrelevant here.

4. Is noise the major factor that makes CMIP projections of East Asian summer monsoon to be non-robust as compared to the top two neutral modes?

Yes, noise (due to the internal variability and model parametric and structural uncertainty) is one of the major reasons. Projection onto much a smaller number of modes, instead of in the original space whose dimension is in the order of 10^4 - 10^6 , can greatly improve the signal noise-ratio for climate change signal detection. See also our response to point 1 of reviewer #1.

5. Given that NM1 and NM2 explain only about 50% of the total variance in precipitation trend, would it be better to include the higher modes. Can adding these additional modes make the sum to be non-robust even though the leading neutral modes are robust.

This is a great idea. As per your suggestion, we now extend the NM-based projection to more modes. Specifically, we following the procedure as follows. Firstly, only the modes of which different models agree on the sign of the projection are used. Secondly, the modes must also satisfy the lenient ToE criterion, that is, “the first year of ensemble mean out”, by year 2100. Lastly, only the modes that contribute more than $1 \text{ mm}^2 \text{ day}^{-2}$ to the variance of the precipitation change are considered for the projection. As a result, we can now extend the NM-based projection to NM10, with NM4 and NM8 excluded (because of their lack of consensus across the ensembles). The comparison between the 2 NM-based and 8 NM-based projection are displayed in Figure R3 below. As you can see, there is no substantial difference between the 2NM-based and 8NM-based projections, except that the magnitude of the latter is somewhat enhanced. We have now included the 8 NM-based projection in a supplementary figure (Figure S6). A discussion has been added in L445-450.

Figure R3. 2 NM-based versus 8 NM-based projections using CESM-SMILEs (a) 2NM-based projection of the JJA precipitation (shading, mm day⁻¹). The variance of the projected precipitation change is 52 mm² day⁻²; (b) same as (a) but based on NM1+NM2+NM3+NM5+NM6+NM7+NM9+NM10. The variance of the projected precipitation change is 78% mm² day⁻².

6. I am not convinced that the model consensus of NMs-based projections can be treated as the consensus for the future projections of East Asian summer monsoon. As pointed in line 286, the results/inferences are constrained by the assumption that the leading neutral modes dominate the total response. More work needs to be done to demonstrate this.

The assumption that “these leading neutral modes dominate the total response” is quoted out of context. This assumption is made only *hypothetically* to deduce the statement that follows: “this regional pattern of summer precipitation change shown in Figure 5a would be something expectable before the end of the century”. Indeed, we agree that more work will be needed to convincingly demonstrate this. This study here is only the first step to set on the path toward more informative and more robust climate change projection over Asian monsoon regions. As we discuss in the last section of the paper, more effort should be dedicated to building consensus on the modes, especially higher modes across models and between models and observations. During the revision, we also realize that a further next step could be to identify possible emergent constraints on the projections of the NMs, once similar neutral modes would be identified for more climate models. For instance, one may make use of the singular values or the pattern correlations of the NM patterns to construct possible emergent constraints. We add the point in L513-516:

“With the NMs identified for multiple models, potential emergent constraints may be more readily established from the NM perspective, with a hope to narrow the spread of the magnitude of the projections onto the leading modes.”

Minor comments :

Fig1 and in other figs too, the colors on the red spectra (1b and 1c) are confusing. It's darker in the middle and is lighter on either side, and it's somewhat difficult to distinguish the red shades. Perhaps use a different colorbar like in Fig 3a for example.

Thanks for pointing this out. We have modified the color scheme for Figure 1 and others to be more intuitive.

Fig1a : define what the gray land regions represent. Also, it might be worth mentioning that the colorbars are in log scale.

Done as suggested.

It might be a good idea to show a similar analysis as in Fig2 for the total precipitation projections, perhaps in supporting information.

This is a very good point. We have now included the conventional multi-model ensemble mean projection for the full precipitation over the Asia-Pacific domain, in comparison with the NM-based projections. The result is reported in the new Figure 5. As you can see that the conventional ensemble projections are quite distinct in both pattern and magnitude compared to the NM-based projection, regardless of whether CESM-SMILEs or CMIP6 model simulations are used.

L200: NM2 index

Fixed.

L325: Ref number for Lu et al is 22.

Fixed.

Reviewers' comments:

Reviewer #1 (Remarks to the Author):

I appreciate the changes made to the manuscript. I am still not convinced of the utility of this approach, but I am ok with that being something that is sorted out in the literature. For the record, though, one thing is to use a fingerprint to detect external forcings in observations, another is to use it on model data that we know is forced. The Hasselmann quote brings up the length of record for a reason! Still, the authors think of this as a first step into a line of research that might bring more fruit later on and I am happy to wait and see.

I do suggest, though, that the authors at least acknowledge the issues with linearity and the use of Green's functions that have emerged from the Green Function MIP (forthcoming paper by Jonah Bloch-Johnson, but also see <https://nadirjeevanjee.com/papers/22mse500.pdf>).

One small curiosity: line 363: are you running positive and negative q-flux perturbations? What are the pairs?

Style: I would often have made different choices about article use. Maybe one more read by a native US speaker would be useful (I am not one, so I don't trust that I am 100% correct on this one either).

Reviewer #2 (Remarks to the Author):

The authors have addressed some of my concerns in this revision. But the main issue still remains, on the use of neutral mode projections to explain the changes in East Asian monsoon. The authors clarify that they don't treat the leading neutral modes as the proxy for the total precipitation change. If it's not representative of the total change or at least doesn't explain a major portion of it, then I fail to see the practical relevance of this work. The variability of the leading modes with climate change would then be an interesting theoretical pursuit but is perhaps not the one suited for Nature Communications.

Here are my major concerns:

1. The authors cite Hasselmann's argument on pattern filtering techniques to justify the NM-based projections. But my main concern is not on the pattern filtering method but rather on the use of neutral modes as the pattern for future change. This needs to be clearly justified.
2. The CMIP6 precipitation change is about an order of magnitude weaker than the NM-projections. The authors explain this being due to cancellation of random biases in the CMIP6 projections, but that needs to be shown or at least cite previous works (if available) to justify it. It's difficult to see the relevance when they are an order of magnitude different and there are also large spatial differences.
3. If the leading modes are able to explain a major portion of the total change, then I would be convinced of their utility. But I am not convinced that the neutral modes represent the "trustworthy" component of the total projections, and should be relied on rather than the CMIP6 projections. That is like arguing the changes associated with the thermodynamic component is more reliable as the total response has additional biases due to dynamic changes. In the end, it's the total CMIP6 change that should be more relevant than just the thermodynamic response, even though the biases in the total projections are larger than the thermodynamic biases.
4. To be clear, I'm not casting apprehensions on the NM-projection method. I'm just not convinced of using the leading modes to talk about climate change. Without that link to climate change, it would perhaps be more of a mathematical tool and might be suited for another journal.

Reviewer #1

1. I appreciate the changes made to the manuscript. I am still not convinced of the utility of the approach, but I am ok with that being something that is sorted out in the literature. For the record, though, one thing is to use a fingerprint to detect external forcings in the observation, another is to use it on model data that we know is forced. The Hasselmann quote brings up the length of record for a reason! Still, the authors think of this as a first step into a line of research that might bring more fruit later on and I am happy to wait and see. I do suggest, though, that the authors at least acknowledge the issue with linearity and the use of Green's functions that have emerged from the Green's Function MIP (forthcoming paper by John Bloch-Johnson, but also see [/https://nadirjeevanjee.com/papers/22mse500.pdf](https://nadirjeevanjee.com/papers/22mse500.pdf))

We thank the reviewer for providing further comments on our manuscript. As the reviewer noted, in climate change detection, the limited length of record is an important challenge motivating the use of the fingerprint approach (or spatial filtering). For climate change projection, our proposed use of neutral modes as a form of spatial filtering is motivated by the large model uncertainty and internal variability that limit our ability to credible future projections. We have made more explicit these contrasting challenges and the motivations of using fingerprinting and neutral modes in climate change detection and climate change projection, respectively.

We also thank the reviewer for bringing up the issue of nonlinearity. We are also acutely aware of nonlinearity in the model used for the extraction of the neutral modes. For example, in our earlier paper (<https://doi.org/10.1038/s41612-020-0112-6>), we proposed an approach to dissect the total temperature response into a linear component and a quadratic nonlinear component (referred to as the symmetric and asymmetric components in the paper).

As per your suggestion, we make note of the issue of nonlinearity in the revised manuscript, citing our own work (Lu et al, 2020) and the forthcoming paper by John Bloch-Johnson on ESS open Archive. In addition, we recently completed another set of Green's function experiments using fully coupled CESM at three-degree resolution. In the presence of ocean dynamics, the nonlinearity is somewhat muted, suggesting that fully coupled climate models may be more linear than slab-coupled climate models and hence the neutral modes of the linear (symmetric) Green's function response can be useful fingerprints for understanding the changes in the fully coupled model.

2. One small curiosity: line 363: are you running positive and negative q-flux perturbations? Why are the pairs?

Yes, we did perform perturbation experiments with opposite q-flux anomalies for each patch. The purpose is to use the difference between the pair of experiments divided by 2 to get the linear/symmetric response; and the neutral modes are the modes of the linear response only.

This is important because the nonlinear response may differ substantially between a slab-coupled system and a fully coupled system (say, the hysteresis of AMOC in the latter system).

3. Style: I would often have made different choices about article use. Maybe one more read by a native US speaker would be useful.

Thanks for the suggestion. We have asked our colleagues/collaborators who are native English speakers to go through the manuscript and fine-tune the written language.

Reviewer #2

1. The authors have addressed some of my concern in this revision. But the main issue still remains, on the use of neutral mode projections to explain the changes in East Asian monsoon. The author clarify that they don't treat leading neutral mode as the proxy for the total precipitation change. If it is not representative of the total change or at least doesn't explain a major portion of it, then I fail to see the practical relevance of the work. The variability of the leading modes with climate change would then be an interesting pursuit but perhaps not the one suited for Nature Communications.

We thank the reviewer for providing further comments on our manuscript, which pushes us to further clarify the value of using neutral modes in climate change projections. We interpret "proxy" as attempting to use the leading neutral modes (NMs) to approximate the multi-model ensemble mean (MEM) CMIP response. If this is what the reviewer means by "proxy", then we emphasize that it isn't our intention to use the changes in the leading NMs as proxy for the total precipitation change often estimated using MEM. We note that the latter often has small magnitudes due to the *cancelation* across models that do not agree on the sign and magnitude of precipitation change. The NM-based projection aims to isolate the robust climate change signals that do not suffer from the cancelation problem because models tend to have more agreement on the changes in the leading NMs. Although the changes in the leading NMs do not represent the full climate change signals, the NM approach is useful because: (1) it allows us to stratify the full climate change response into components that are robust and not robust; (2) it enables investigation to understand the processes driving the robust changes; (3) it allows us to communicate our understanding of the robust changes and highlight the need and potentially the approaches to reduce uncertainty in the non-robust changes; and (4) it has potential to gradually asymptote to the true full response as climate models improve and consensus is built on more modes.

To further support our statements, we present additional evidence and analysis as elaborated below.

(a) Multi-model climate projections have large uncertainty

Figure R1 below adopted from Neelin et al. (2006) shows little agreement on the tropical JJA rainfall trend over 1979-2099 across the 20 CMIP models examined. Since models have little

agreement on the dry/wet trends (panel b), the MMEM change would be muted and not very useful. Hence climate change projections are often presented not only using MMEM but also including stippling to show the model agreement. Our NM-based effort for climate projection aims to elucidate the robust components despite a highly uncertain total precipitation response.

Figure R1 (a) Precipitation trend for JJA of the multi-model ensemble median from 1979 to 2099. Shading indicates 99% significance by the Spearman-rho test. The black line gives the 4mmday₋₁ contour from the median climatology (1900–1999 average) of the models to indicate a typical boundary of the convection zones. (b) Model agreement on the predicted local precipitation trend from 1979 to 2099 for JJA. The number of models at each location that agree on a dry trend or a wet trend exceeding 99% significance and exceeding a minimum amplitude change (20% of the median climatology per century) is given by the brown or green color bars, respectively. Only regions with five or more of the 10 models agreeing are shaded.

(b) Neutral modes can recover the total precipitation response of a “perfect” model

Although the changes in the leading NMs do not represent the total precipitation response, it is important to note that the NMs can recover the total precipitation response of an individual model when more modes are included. To demonstrate this point, we conducted NM-based projections for a “perfect model” scenario, using increasing number of NMs for the trend simulated in the CESM-LENS2 ensemble, which has ~100 members so the ensemble mean change is representative of the “true” climate change response to anthropogenic forcing in this model. In Figure R2 below, as more modes are included in the projection, the result asymptotes to the “true” forced change estimated by the CESM-LENS ensemble mean. In this perfect model example, the projection based on 5 modes (upper left panel) does not differ significantly from the ensemble mean (lower right panel), and the difference between them can provide useful insights on regional processes that govern the changes in the higher modes such as the

topographic signal associated with the Tibetan Plateau to the west of the domain. Stratifying the total change by the modes allow us to develop understanding of different processes governing the changes at different scales.

Figure R2 Approaching the ‘true’ CESM-LES2 mean trend using NM-based projection The NM-based projection can approach the ensemble mean trend of CESM-LES2 as a larger number of NMs is used for projection. At the top-right corner of each panel, the fraction of the co-variance with the ensemble mean trend is labeled. Denoting the ensemble mean trend pattern as δP and the NM-based projections pattern as P_n , with n indicating number of NMs included for the projection, the fraction of the co-variance with P pattern is defined as $\frac{\langle \delta P_n, \delta P \rangle}{\langle \delta P, \delta P \rangle}$, where $\langle \cdot \rangle$ indicates inner product.

(c) Neutral modes can recover the multi-model mean precipitation response

Although the changes in the leading NMs of the multi-model ensemble do not represent the MMEM precipitation response, it is important to note that the NMs can recover the MMEM precipitation response when more modes are included. To clearly demonstrate this, we sequentially increase the number of NMs included for the NM-based projections and the result is reported in Figure R3 below. As more and more modes are included, the resultant pattern of projection has weaker and weaker magnitude, as anticipated from cancellation due to model disagreement. In panel (f) which uses 50 NMs, the NM based projection is very much approaching the MMEM response (cf. Figure 5b in the revised manuscript).

Figure R3 The NM-based multi-model projections with increasing number of NMs included. When a large enough number of modes is considered, the resultant projection can gradually approach the projection using conventional multi-model ensemble mean (cf. Figure 5b in main text).

Let's examine how the NM-based projection captures the fractional variance of the ensemble mean projection of CESM-SMILES (denoted as δP) as the number of modes is increased (blue line in Figure R4), together with how the averaged inverse signal-to-noise ratio ($1/\text{SNR}$) over the modes used for projection (red line) varies with the number of modes. $\text{SNR}=1$ indicates that noise and signal on average are at similar magnitude after projecting on to the neutral modes. For instance, using 10 NMs results in a $1/\text{SNR}=1.7$, meaning that the noise is 70% larger than the signal.

From Figure R4, one can clearly see a tradeoff between the variance explained and the confidence in the projection. While more modes allow the NM-based projections to approach the total response estimated by the conventional MMEM, this is achieved at the cost of increasing model spread-to-signal ratio (MSSR, red line, calculated as the ratio of the variance of the inter-model spread to the variance of the multi-model ensemble mean) or increasing uncertainty. Also shown in Figure R4 is the fraction of variance of the CESM-LENS2 ensemble mean trend explained by the NM-based projection (blue dashed line), in which CESM-LENS2 is treated as the true climate and other SMILES model simulations as numerical models. The non-monotonic variation in this curve (compared to the solid blue curve) lays bare the difficulty of

(imperfect) models in capturing the total forced climate change signal. The difference between the two blue curves is attributable to the *cancellation* due to inter-model spread or model uncertainty (if all the SMILEs models behaved like CESM-LENS2 (the “perfect” model), the two blue curves would overlap). Importantly, the two blue curves do overlap if only the first few NM modes are used in the projections, demonstrating broad model agreements on the projected changes in the first few NM modes. The peak point on the blue dashed curve could possibly be used as a criterion for choosing the number of modes to include in the NM-based projections, but this peak point can vary with the number of models and which ensemble is treated as the perfect model. A more empirical approach is taken in the revision to select the confident modes for climate projection (see Methods 3. Criteria for choosing NMs for precipitation projection).

In summary, given the modest fidelity of the current generation of climate models, one can only establish some confidence in a small number of modes representing larger-scale patterns. Using more modes (e.g., > 5 as shown in Figure R4) reduces the signal-to-noise ratio due to model disagreement in the higher modes. While using MMEM to estimate the precipitation response reduces the impact of model uncertainty on the projection, it does not increase our confidence in the model projection if model agreement on the projection is low. The NM-based approach offers a useful alternative in understanding and presenting climate change response by focusing on the robust components as delineated by the leading neutral modes of the dynamical system.

Figure R4 A trade-off between number of NMs and confident projection with high MSSR. Blue line: The fraction of the variance of the CESM-SMILEs ensemble mean precipitation change (i.e., δP) explained by the NM-based projections as the number of NMs included increases (y-axis=1 means 100% variance explained). Blue dashed line: The fraction of variance of the CESM-LENS2 ensemble precipitation change (i.e., δP_{LENS2}) explained by the NM-based projections using other SMILEs model ensembles. Red line: The MSSR for each NM-based projection computed as the ratio of the variance of the inter-model spread to the variance of the multi-model mean signal (y-axis=1 means noise level is as high as the signal).

2. The CMIP6 projection change is about an order of magnitude weaker than the NM-projections. The authors explain this being due to cancellation of random biases in the CMIP6 projections, but that needs to be shown or at least cite previous works (if available) to justify it. It’s difficult to see the relevance when they are an order of magnitude difference and there are also large spatial difference.

We hope to have addressed this comment in our more comprehensive response to the first comment elaborated above. More specifically, previous studies have highlighted the lack of model agreement on the wet/dry trends in model projections (Figure R1) that can result in a weak precipitation response as estimated by MMEM. Model disagreement (and hence cancellation of model responses) is further demonstrated by Figures R3 and R4 using CESM-LENS2 as the “perfect” model or true climate and the other SMILEs models as numerical models with biases and uncertainties.

3. If the leading modes are able to explain a major portion of the total change, then I would be convinced of their utility. But I am not convinced that the neutral modes represent the “trustworthy” component of the total projections and **should be relied on rather than the CMIP6 projections**. That is like arguing the changes associated with the thermodynamic component is more reliable as the total response has additional biases due to dynamic changes. In the end, it’s the total CMIP6 change that should be more relevant than the thermodynamic response, even though the biases in the total projections are larger than the thermodynamic biases.

To be clear, I’m not casting apprehensions on the NM-projection method. I’m just not convinced of using the leading modes to talk about climate change. Without that link to climate change, it would perhaps be more of a mathematical tool and might be suited for another journal.

We hope our detailed response to comment #1 helps clear your concern regarding the relevance of our NM-based approach for climate projection. More specifically, we have demonstrated that neutral modes can recover the total precipitation response of a single model as well as the multi-model ensemble mean response as more modes are included in the projection. However, including more modes reduces the signal-to-noise ratio (Figure R4), reflecting the large uncertainty in the total precipitation response estimated using MMEM. And Figure R4 further support the model agreement on the first few modes, and hence their robustness in practice, besides the theoretical underpinning of the neutral modes as the intrinsic modes of the dynamical systems and that the leading modes resemble the EOFs derived from observations.

We are not arguing that the changes in the leading modes represent the total change, and we recognize the importance of projecting the total change to support decision making. We present the NM-based projection as a useful approach to stratify the total response into robust and non-robust components that allows us to understand the processes responsible for these components and devote efforts to improve confidence in the non-robust components towards the ultimate goal of credible projections of the total response. In this sense, the mode-based approach complements the traditional approach of estimating climate change response using multi-model ensemble mean and directs attention to the gaps in understanding and projecting changes in the higher modes to achieve credible projection of the total response.

REVIEWERS' COMMENTS

Reviewer #2 (Remarks to the Author):

I appreciate the additional changes made by the authors. They have addressed my major concerns and have done a good job of revising this manuscript. I'm not entirely convinced of the use of leading neutral modes to represent the "robust" changes in comparison to the total change. I guess my main concern is on precision vs accuracy, in that, leading modes are more robust (or precise) but may not be an accurate "metric" to study climate change. Perhaps, authors can add a few lines to discuss this or may be even just acknowledge it as a limitation/caveat. But nevertheless, I agree with the other reviewer that this manuscript can be published now and other issues can be sorted out in future works.

Reviewer #2

I appreciate the additional changes made by the authors. They have addressed my major concerns and have done a good job of revising this manuscript. I'm not entirely convinced of the use of leading neutral modes to represent the "robust" changes in comparison to the total change. I guess my main concern is on precision vs accuracy, in that, leading modes are more robust (or precise) but may not be an accurate "metric" to study climate change. Perhaps, authors can add few lines to discuss this or may be even just acknowledge it as a limitation/caveat. But nevertheless, I agree with the other reviewer that this manuscript can be published now and other issues can be sorted out in future works.

We appreciate the reviewer's perceptive comment of "precision v accuracy". Indeed, our mode-based approach can give more robust (precise) regional projection than the conventional point-wise projection, but we cannot argue that our approach is more accurate as of now. We have added a brief discussion in the text (L365-368) about the importance of discerning the two different metrics for climate projection as follows.

"It is also important to note the difference between robustness and accuracy of the regional climate projection: the NM-based approach can bring out the robust aspect of the regional response, but it does not necessarily lead to more accurate regional climate projection. The latter will be predicated on building agreements on higher and higher orders of the NMs across climate models."